# Spotting what's important: Priority areas, connectivity, and conservation of the Northern Tiger Cat (*Leopardus tigrinus*) in Colombia

**José F. González-Maya**[1,2]*, **Diego A. Zárrate-Charry**[1,3], **Andrés Arias-Alzate**[4], **Leonardo Lemus-Mejía**[1], **Angela P. Hurtado-Moreno**[1], **Magda Gissella Vargas-Gómez**[5], **Teresa Andrea Cárdenas**[5], **Victor Mallarino**[6], **Jan Schipper**[7]

1 Proyecto de Conservación de Aguas y Tierras–ProCAT Colombia, Bogotá, Colombia, 2 Departamento de Ciencias Ambientales, División de Ciencias Biológicas y de la Salud, Universidad Autónoma Metropolitana Unidad Lerma, Lerma de Villada, Estado de México, México, 3 WWF Colombia, Bogotá, Colombia, 4 Facultad de Ciencias y Biotecnología, Universidad CES, Medellín, Colombia, 5 Grupo Energía Bogotá S. A. E.S.P., Bogotá D.C., Colombia, 6 Independent, Bogotá D.C., Colombia, 7 Arizona Center for Nature Conservation/Phoenix Zoo, Phoenix, Arizona, United States of America

* jfgonzalezmaya@gmail.com

**Data Availability Statement:** Data about the species are within the manuscript and its Supporting Information files. Climatic data is available from WorldClim (https://www.worldclim.

## Abstract

*Leopardus tigrinus* is among the least known carnivore species in the Neotropics, including considerable taxonomic uncertainty. Here we model the distribution, connectivity and overlap with existing conservation areas for the species in Colombia. Using a Species Distribution Modeling approach, we estimated current potential range of the species in Colombia and identified potential habitat blocks remaining in the country. In addition, we designed a connectivity network across the available cores, using a circuit theory approach, to evaluate habitat linkage. Finally, we defined a prioritization scheme for the remaining habitat cores and assessed the level of coverage of protected areas for the country. *L. tigrinus* is potentially present across the three Andean branches of Colombia, with still considerable continuous habitat cores, mostly located on the eastern and central Andean ranges. Most habitat cores are theoretically connected, but nearly 15% are isolated. Priority areas were located across the eastern and central ranges, but with very significant and promising cores in the northern eastern and western ranges. Current level of protection indicates nearly 30% of the range is "protected", but only about 25% is under national strict protected areas. Evolution of this coverage showed some periods of significant increase but interestingly the number of cores grew at a faster rate than overall proportion protected, likely indicating numerous discontinuous fragments, and not contiguous functional landscapes. This represents the most updated assessment of the distribution and conservation status for the species in Colombia, and indicates the numerous conservation opportunities, especially in most populated areas of the country. We found unique business environmental passive's opportunities, including compensation and development potential, which are becoming more available in the country.

org). Protected areas information is available from the Registro Único Nacional de Áreas Protegidas (https://runap.parquesnacionales.gov.co/).

**Funding:** This project was jointly funded by ProCAT Colombia and Grupo de Energía de Bogotá S.A E.S.P through mutual agreement. The funders had no role in study design, data collection and analysis, decision to publish, or preparation of the manuscript.

**Competing interests:** The authors have declared that no competing interests exist.

## Introduction

Effective species conservation requires solid scientific information to support decision- and policy-making [1]. Basic information such as distribution, abundance and threats are among the most important aspects needed for effective conservation planning [2, 3], especially when such a large number of species are under threat [4]. Despite their charismatic nature, important role in ecosystem balance and general ecological importance, most carnivores lack even the most basic information [5–7], undermining effective conservation actions, especially given their particular sensitivity to the most pressing threats [8].

Despite the fact that massive efforts have concentrated on a handful of "mega-charismatic" carnivore species worldwide [6], small carnivores have seemingly been overlooked [5, 9]. In fact, some small carnivore species are considered among the least known among mammalian fauna in many countries [5, 10]. Even for wild felids, one of the groups which have received more attention [11], the smaller species are the least understood while at the same time being among the most threatened [11–13].

This is certainly the case for the Northern Tigrina or Northern Tiger Cat, *Leopardus tigrinus*, one of the smallest felids with a wide distribution in the Americas, and perhaps the least understood spotted cat known in the continent [14–16]. Important gaps in information about the species include certainty of its taxonomic status [17–19], as well as even basic knowledge of its distribution, ecology and conservation status [14, 15, 19]; however, significant progress have provided new insights into the group´s taxonomy [16–18]. With a considerably large range, likely from Costa Rica down to Bolivia and Peru, and East across Venezuela and the Guianas [17], still taxonomic uncertainty does not allow precise distribution assessments, varying according to different authors and taxonomic approaches [16–18]. Mostly unknown, both ecologically and taxonomically, almost all information about Northern Tiger Cat across its range countries is restricted to just occasional records and sporadic observations, but with no systematic data on any other aspects of its ecology and especially conservation status [14, 20–23].

Specifically, in Colombia the species is only known from locality records and observations, with most of the literature focused on distribution and confirmed localities [14, 22, 24], with very few National scale systematic approaches to its distribution and conservation status [14]. Also, in Colombia the species is mostly distributed at higher elevations in the Andes, above 1,500 m asl [14], which is also one of the most transformed and heavily populated regions in the country [25]. Given the considerable pressure over Andean ecosystems in general [26–28], and the specific pressure that such transformation exerts over the species [15], understanding the species distribution and conservation status represents an urgent need as its remaining habitat slowly disappears. Furthermore, given the considerable human development pressure on the remaining habitat for the species, the Northern Tiger Cat has become a predominant species challenging many development and infrastructure projects in the country, necessitating better information that can help informing and plan for its conservation. This could represent a unique conservation opportunity for the species, especially considering the substantial compensation requirements of companies interested in these types of projects. To provide baseline information that can be incorporated in future conservation planning, including compensation and licensing of future development projects, herein we assess the distribution and conservation status of the species in the country. Specifically, i) we assessed the distribution of the species at a national scale, ii) illustrate potential connectivity networks across remnant habitats, iii) selected and categorized discrete priority remaining habitats, and iv) assessed the conservation status of the priority habitats.

## Materials and methods

### Study area

Our study focused on the known distribution range of Northern Tiger Cat in Colombia [14], which encompasses the entire terrestrial portion above 2000 m asl of the country. Colombia is located in northwestern South America, bordering with Brazil, Venezuela, Peru, Ecuador and Panama; it is located on the northernmost part of the continent and serves as the only terrestrial limit with Central America. Given the focus of our study, it is worth mentioning that Colombia is also the northernmost part of the Andes where the mountain range splits into three branches (i.e., western, central, and eastern ranges), where they reach elevations over 5000 m. The unique features of the northern Andes represent an interesting case study in terms of evolution and biogeography; the divided mountain range creates unique conditions for a large variety of life-forms with many species ranging on the three branches and some restricted to either of them [29]. Although our study focuses on the higher parts of the Andes, some isolated ranges are also included in the analyses, including the Macarena, Perijá, San Lucas and Sierra Nevada de Santa Marta. These are all independent mountain ranges for the Andes but also reaching elevations where the species is known or thought to occur.

### Methods

**Species' records.** Our species distribution model (SDM) was based on the most updated compilation of confirmed available records for the species, including a thorough literature review [14, 17, 22, 30–37], and consulting all available national and international databases (i.e., SIB Colombia, GBIF, VertNet [38, 39]) covering a temporal range between 1951 and 2019.

Once a final databased was compiled, we discarded all records without specific locality information or coordinates and then used three criteria filtering for data quality control–reducing the available data to only confirmed locations (S1 Table). We generated a categorization scheme for each record based on three criteria: source, evidence and geographic precision; precision was evaluated by comparing the coincidence between locality details reported in the source and the national cartography of Colombia (Table 1; [40]). Each criterion showed

**Table 1. Criteria used for classifying reliability of records for modeling the potential distribution of *L. tigrinus* in Colombia.**

| Criteria | Attributes | Reliability |
|---|---|---|
| Evidence | Preserved specimens | High |
| | Machine observation/Photographic record | High |
| | Material sample | Medium |
| | Human observation | Low |
| | No data | Low |
| Source | Peer reviewed article | High |
| | Expert validated record | High |
| | Museum | Medium |
| | GBIF | Medium |
| | Technical information | Low |
| Geographic precision | Department and Municipality | High |
| | Department | Medium |
| | Municipality | Medium |
| | None | Low |

different attributes with an assigned reliability category (high, medium, low) that were latter cross referenced for each record (Table 1). We then applied a moderate filter, in which any record with one category ranked as low would be excluded from the final model [41, 42]; we considered this filter as moderate given that we did not use only those categorized as high in the three criteria as in previous efforts [41–43]. Finally, we eliminated duplicates and used a spatial filter which eliminated any redundant records on a 1 km$^2$ grid [44].

**Distribution modelling and potential habitats.** For the assessment of the species potential distribution, we generated a SDM based on an ecological niche modeling approach [45] using the maximum entropy algorithm [46]. To define the calibration area of the SDM we obtained an updated polygon layer of the world´s ecoregions [47] as the Mobility area (M) for the species [48] and selected those ecoregions that had at least one record of the species after filtering and we narrowed-down all the inferences to the extent of those ecoregions [49]; all ecoregions selected correspond to those ecoregions associated with the high Andean ecosystems (S1 Fig). We then obtained bioclimatic variables from Worldclim 1.4 [44], including those that represented best the variation for a species such as Northern Tiger Cat, and that have proved useful for modeling carnivore distributions [50, 51]. Specifically, we used seven bioclimatic variables: Mean Annual Temperature (Bio1), Diurnal Mean range (Bio2), Temperature seasonality (Bio4), Annual precipitation (Bio12) and Precipitation seasonality (Bio14), Precipitation of the wettest month (Bio13) and Precipitation of the driest month (Bio14). These last two variables have shown a better representation of the ENSO (El Niño-Southern Oscillation) effects [52]. Finally, we included elevation [40], given its potential influence on species distribution models, especially for the Andean region [49]. We then analyzed potential correlation between these variables across our calibration areas using a Spearman correlation ranks test and the jackknife test based on the maximum entropy model to select those with correlation scores under 70% and the highest permutation values [41].

We used a maximum entropy approach available through the software Wallace 1.0.6.2 [53], which include different R-based packages, including Maxent [54] and ENMeval [55]. Maximum entropy analysis is the most widely used algorithm for quantifying the relationship between the presence of a species and environmental variables [56, 57]. The records database was divided into calibration and testing dataset following Cobos et al. [58]. For each set of variables, we created candidate models (based on 10 replicates by bootstrapping and selecting the logistic output) using the calibration occurrences, 17 different Regularization Multiplier (RM) values (0.1–0.9 at intervals of 0.1, 1–6 at intervals of one, and 8, and 10), 29 possible Feature Class (FC) combinations of five feature classes (linear = l, quadratic = q, product = p, threshold = t, and hinge = h), with 10,000 background points. These candidate models were evaluated and selected using the testing dataset, based on statistical significance (Partial ROC, with 500 iterations and 50% data for bootstrapping), omission rate (OR ≤5%), and model complexity with the Akaike Information Criterion corrected for small sample sizes (AICc) [58]. The resulting model was then transformed into binary (i.e., presence-absence), following a conservative threshold, using the 10 percentile training presence logistic threshold value [59], averaged across the 10 replicates.

**Selection of available suitable habitats.** Once a binary map was obtained from the best model, we evaluated the different types of vegetation cover included in the suitable areas according to available information on species' use [24, 42]. All coverage catalogued as natural by the national cartography dataset [60] and with existing evidence of use by the species from literature and the distribution records, was selected within the binary map [42]. Once all coverages were extracted, we defined an area threshold for selecting those remaining patches that could be used by the species as core habitats, based on a conservative value estimated from the maximum home range reported for the species of 17 km$^2$. The selection of such threshold

aims at prioritizing those core habitat patches that ensure viability of at least one individual considering the inherent uncertainty of the home range size for the species and the base- data obtained from a different but closely related species (reported values range from 0.9 to 17,1 km$^2$; [61]).

**Connectivity networks definition.** Once all available remaining habitats were identified, we designed an ecological connectivity network analysis across the recognized patches based on a circuit theory approach [62, 63]. The ecological connectivity network is based on the resistance that certain landscape features exert over the potential dispersibility of a given species between its core habitats [62, 63]. We consider as core habitats the recognized habitat patches remaining within the suitable areas identified in the species' distribution model. The resistance layer was generated based on the available "human footprint index" developed for Colombia [64]. This human footprint approach includes land uses, rural population density, distance to human settlements and roads, a general fragmentation index, a comparative biomass index compared to the original, and time of intervention [64]. These variables represent multiple levels of human influence over landscapes, including current land use, which can be also used as a proxy to potential barriers (i.e., resistance) to movement across fragmented landscapes [64]. Our assumption is that without knowing the species-specific resistance values that multiple interacting variables can exert over species dispersal, a standardized and weighted human-influence index can validly reflect the resistance of such influence over species movement (especially on highly intervened landscapes such as the Colombian Andean region) [65]. We then used the 2015 Human spatial footprint for Colombia, comparable to our coverage layer for 2016 [60] and used a linear transformation to turn the footprint into a resistance layer by rescaling it to a 1–100 standard scale [66]. The connectivity network was then confined to a 15 km dispersal distance in order to identify corridors that pertain to the species potential movements and thus avoiding overestimations [42]. This value represents a potential dispersal distance of an individual with a home range of 4.5 km$^2$, which represent the median value of all reported home ranges for the species [42, 61, 67]. This method has been previously used for similar purposes in Colombia and the region [31, 41, 68–70]. We then identified the importance of each core habitat by estimating its centrality value (CF_Central value) and the potential current flow for each link within the connectivity network; centrality represents a measure of how important is a linkage for maintaining the overall connectivity network connected, while current flow helps predicting the expected net movement probability of an individual through the linkage [63, 71–73]. We tested the relationship of core habitat size with centrality value using a linear regression and decremental regression analyses (i.e., chained regressions) excluding cores over different size thresholds. We used the adjusted R$^2$ values to find the threshold where the relationship significantly decreases. Our connectivity network was developed using the Linkage Mapper 10.x toolbox [71, 72] available for ArcGIS [74].

**Selection and categorization of priority areas.** Once a network of core suitable habitats and a connectivity network was established, we defined discrete units for each core and estimated the following parameters to stablish a priority scheme for Northern Tiger Cat in Colombia. We defined three attributes of the classification criteria that allowed a weighted ranking of all core areas to define a prioritization scheme according to: i) Size, ii) Mean human footprint and iii) Centrality value. Size was estimated as the total area of each discrete polygon identified as core habitat (log-transformed), mean human footprint as the mean value of the 2015 human footprint overlapping the polygon [64] and centrality (log-transformed) as estimated by the circuit analyses [72]. We then summed up the three categories and estimated a final value for each core which was then qualified, categorized, and standardized generating an ordinal priority from 1 to 5, from the definition of an equal interval of the total range of values,

identifying those cores with the best existing conditions (1) to those severely fragmented or affected by human intervention (5).

**Conservation measures for Northern Tiger Cat populations in Colombia.** Finally, for each core area we estimated several species conservation measures including two parameters and our prioritization scheme. Parameters used are i) number of individuals potentially present on each core and ii) coverage of protected areas. The number of individuals was estimated by extrapolation from known density estimations found elsewhere [15] to the available core habitat areas identified through our models. Density estimations are only available for Brazilian tiger cats [15, 67, 75, 76], so we used all these reports to calculate the minimum number of individuals and the potential variation between estimations; we estimated upper and lower limits and, mean estimation from different sources and distribution of the total number of individuals expected. Specifically, we used complementary estimations, including those considered high, typical and those previously reported for fragmented landscapes according to previous authors [15, 75], and our mean estimate from the existing information. It is necessary to acknowledge the potential bias of our approach given that no specific information exists for the species and for Colombia, and that likely these estimates correspond to *L. guttulus* (Southern Tiger Cat); nevertheless, given this is the best available information, we believe this approach can be informative until better and more specific information becomes available.

For assessing the coverage of protected areas we estimated the overlapping of the remaining core habitats with the national official protected areas layer [77]. First, we estimated the proportion of core areas covered by all protected areas from the national, regional, and local levels reporting the total level of protection over the entire SDM and for each core. We then estimated the proportion of cores completely or partially protected and according to the protection type. Finally, we assessed how the coverage of protected areas have changed over time to understand how such protection has evolved in terms of total covered area and number of cores fragments included.

All maps were created using ArcGIS [74] and we used a global Digital Elevation Model [78] as background for all map figures.

## Results

### Distribution of Northern Tiger Cat in Colombia

We obtained a total of 448 records of the species, of which 212 were considered of high quality and credibility (S1 Table). Our sampling area included 6 ecoregions with previously confirmed presence of the Northern Tiger Cat: Cauca Valley montane forests, Cordillera Oriental montane forests, Eastern Cordillera Real montane forests, Magdalena Valley montane forests, Northern Andean paramo and Northwest Andean montane forests (S1 Fig). Derived from the correlation tests, we eliminated from further analyses Annual Mean Temperature, Precipitation of the Wettest Month and Precipitation Seasonality, which were highly correlated to elevation (Spearman = -0.90), Annual Precipitation (Spearman = 0.77) and Precipitation of the Driest Month (Spearman = -0.94), respectively (S2 Table). Among the selected variables, elevation (64.3%) and temperature seasonality (27.9%), accounted as the most important according to permutation importance, but with Annual Precipitation (6.3%), Precipitation of Driest Month (0.94%) and Mean Diurnal Range (0.45%), also influencing its pattern. As expected, elevation had a positive influence on the species´ model (higher probabilities at higher elevations), together with Annual Precipitation, while Temperature Seasonality, Precipitation of the Driest Month and Mean Diurnal Range showed predominantly negative influence (higher variable values, lower probabilities), although with some interesting changes (S2 Fig). The most informative and best selected models were the LQHP (Linear Quadratic Hinge Product)

feature class and the 2.5 and 2.0 regularization multipliers models (AICc: 4323.54 and 4323.61, respectively), both models indicate good performance with AUC of 0.8902 and 0.8923, respectively (S3 Table). The resulting 10 percentile training presence logistic threshold binary map, now on SDM, had a total of 228817.43 km$^2$ of environmentally suitable areas distributed in the three branches of the Andes in Colombia (Fig 1A). However, given the large-scale human-caused natural cover transformation processes across the distribution of the species, our selection of suitable remaining habitats significantly reduced these available areas for the species (Fig 1B).

After identifying the available natural coverages within these suitable areas, we found the potentially occupied areas to be only 91,209.85 km$^2$ (~39.8% of the SDM), distributed in 183 patches, with a mean (±SD) area of 498.41±2783.58 km$^2$ across the SDM (Fig 2A). This area corresponds to less than 8% of the country area and nearly 40% of the suitable area. We found significant variation among fragments, with many patches covering barely the 17 km$^2$ threshold (155 (84.7%) under 20 km$^2$), with 93% under 1,000 km$^2$ and only one remaining patch covering over 35,000 km$^2$ (Fig 2B). Most remaining habitat patches were concentrated towards the smaller size, with very few close to the largest one (Fig 2B); although, only 12 patches (all over 100 km$^2$) cover together over 76,000 km$^2$, representing over 83% of the of the identified core areas. The largest remaining habitats are located on the southern central and eastern mountain ranges, including the "macizo colombiano", where both ranges divide, followed by the northern portion of the eastern range (Fig 2A).

## Potential connectivity network for Leopardus tigrinus in Colombia

Our ecological network analyses selected those areas with the lowest cost for dispersal between habitat cores according to the lowest cost weighted distance between each core. The resulting network, pruned to the maximum 15 km threshold dispersal value, identified approximately 250 links between habitat cores, varying in terms of Euclidean and length distance variation and potential current flow (*sensu* McRae, Dickson [72]). Corridor links mean (±SD) Euclidean and length distances were 2.52±2.45 and 3.31±2.97 km, respectively, while potential current flow was 504.42±650.66 (Fig 3A). Interestingly, 156 of the total 183 had at least one linkage to other cores and the mean (±SD) number of links between cores was 1.61±0.80, with the largest number reaching 5 for a single core.

Centrality values for all cores also varied considerably, with a mean value of 981.53±1128.85 for the entire network, but with an interesting geographical variation (Fig 3A); overall centrality value was closely related with the size of the core (T = 16.40, p<0.001, R$^2$Adj = 0.60; Fig 3B), indicating both a geographic (location) and spatial influence on the entire network. However, the centrality value variations explained by area (Adjusted R$^2$) tended to decrease towards the smaller cores, indicating significant importance of the largest ones. Additionally, there is a peak for all cores below 2,000 km$^2$ (T = 11.17, p<0.001, R$^2$Adj = 0.42), which rapidly decrease until only cores below 100 km$^2$ are included, where the influence is no longer significant (T = 1.09, p = 0.2782, R$^2$Adj = 0.013; Fig 3B).

## Priority areas for the conservation of Northern Tiger Cat in Colombia

Our approach for priority areas selection identified those cores that represent the best opportunities for conservation given their available area, their contribution to connectivity and, ideally, where human influence is lower. Mean (±SD) area for all cores was 498.41±2791.22 km$^2$, but with a concentration towards the smallest patches, as mentioned above. Mean (±SD) value for centrality, importance for the maintenance of the connectivity network, was 981.53

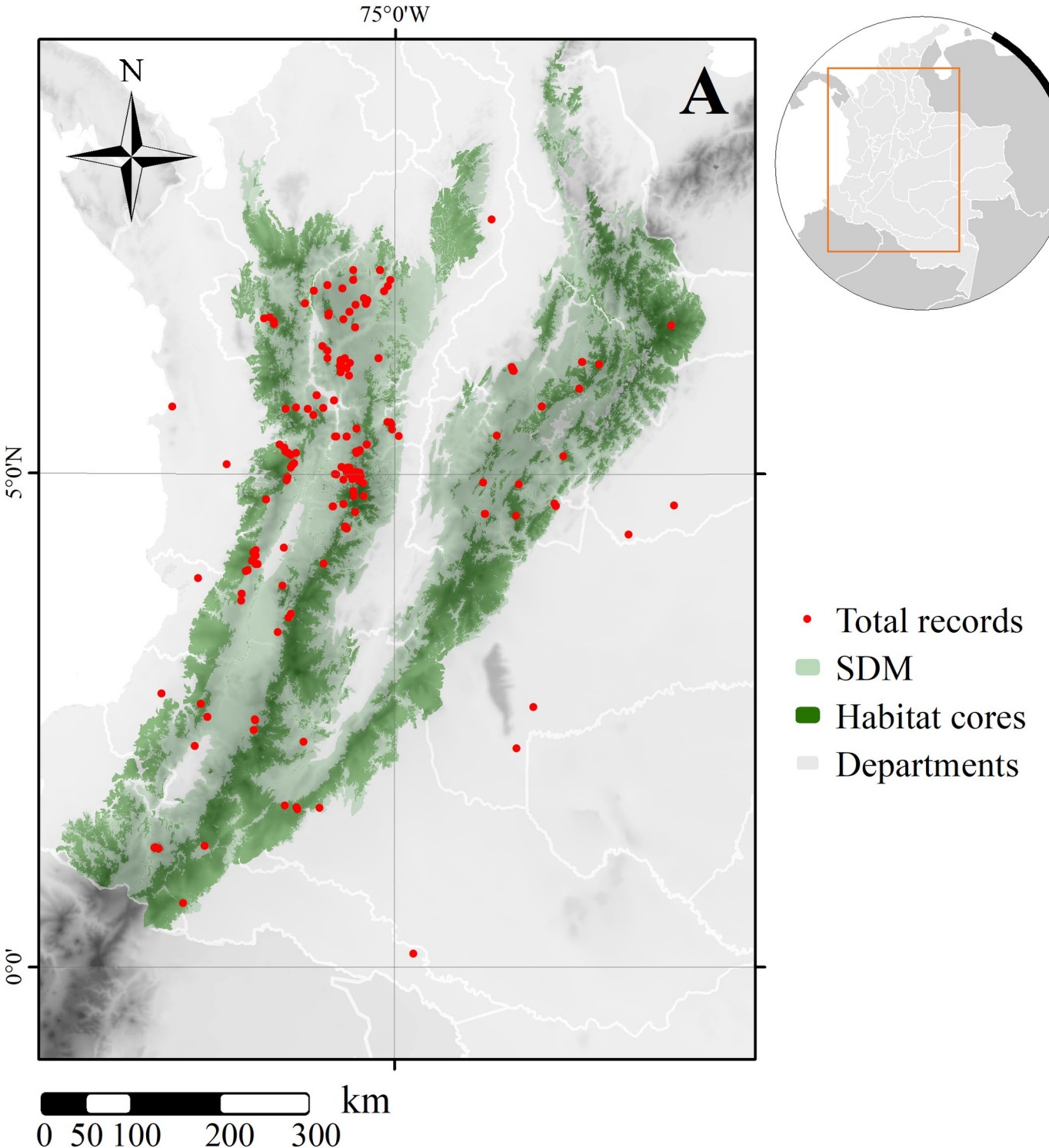

**Fig 1. Distribution of total records obtained for *Leopardus tigrinus* over potential distribution (SDM) in Colombia and remaining habitat cores.** Background shows topography [78], light gray depicts Colombia and white lines define the political and administrative divisions of Colombia (departments). Map created using ArcGIS [75].

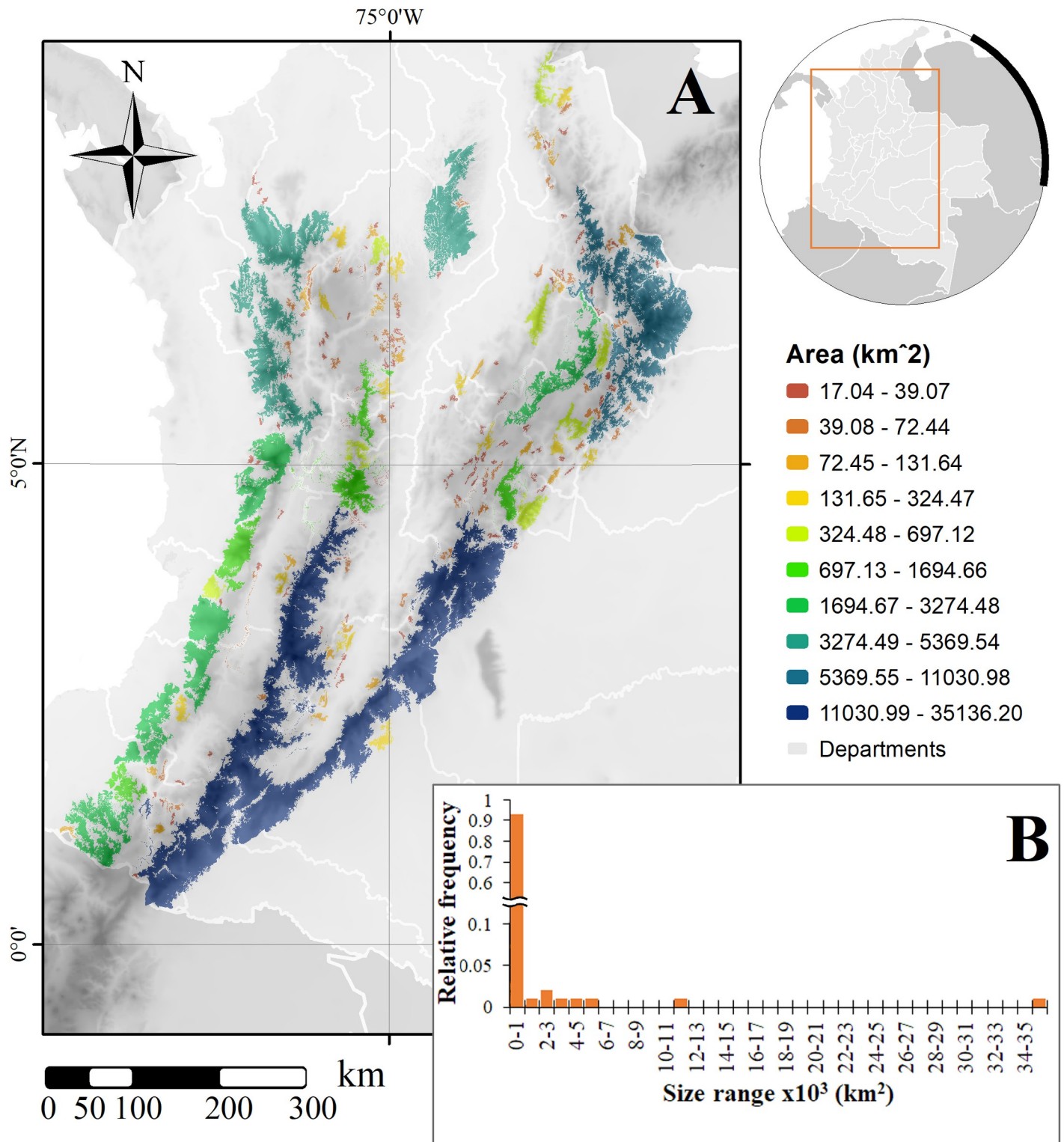

**Fig 2.** (A) Size of available habitat cores for *Leopardus tigrinus* in Colombia and (B) distribution of fragments according to size classes. Background shows topography [79]; map created using ArcGIS [75].

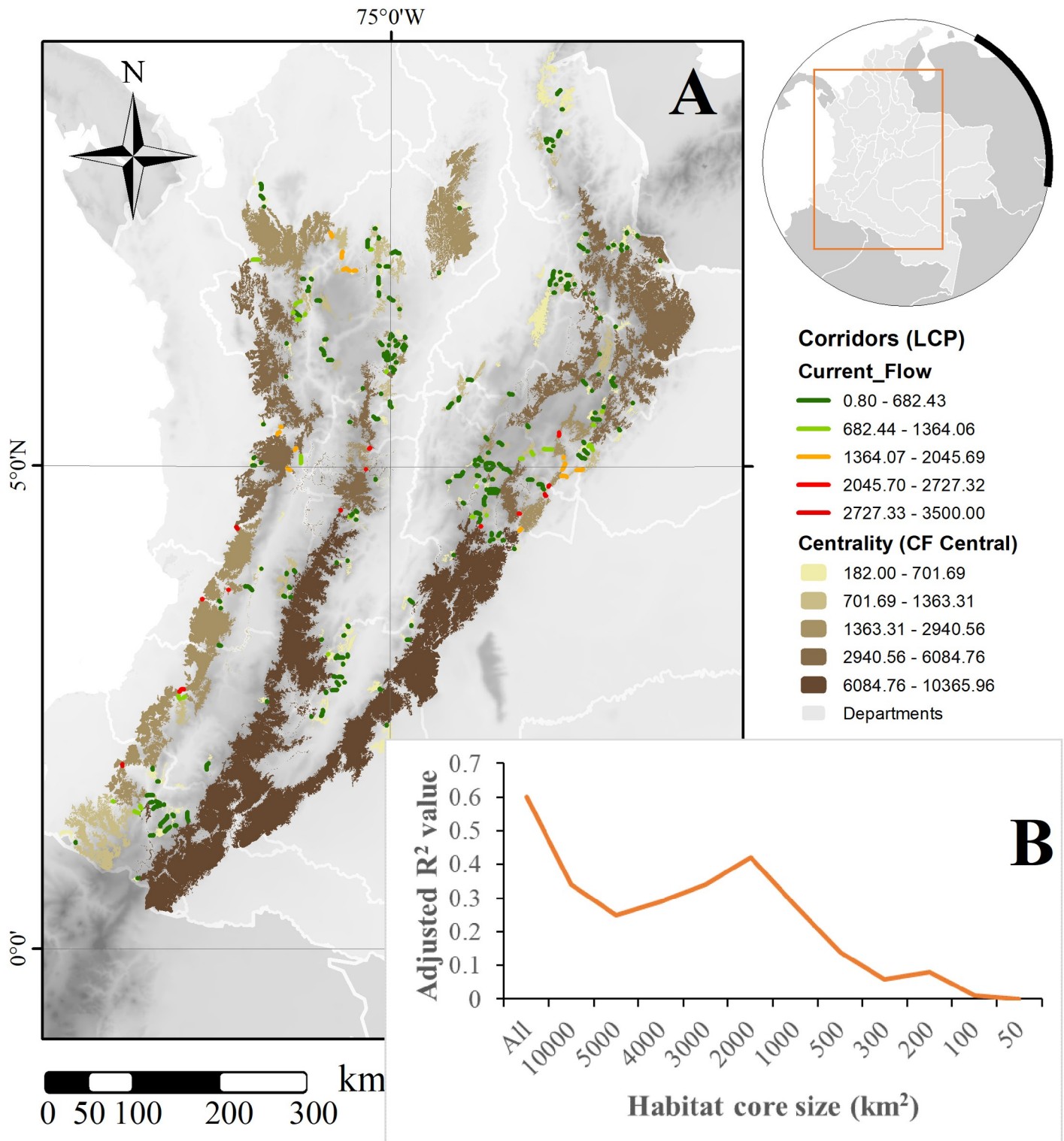

**Fig 3.** Distribution of (A) potential connectivity corridors and centrality values across remaining habitat cores for *L. tigrinus* in Colombia and (B) decremental variation of connectivity importance explained by core area size. Background shows topography [79]; map created using ArcGIS [75].

±1131.96, where the largest patches were also coincident with those that contribute more significantly to overall connectivity (Fig 3A).

In terms of human influence, mean (±SD) index values for all cores was 35.20±19.53, also showing that the largest cores remain with lower human influence (Fig 4B).

After weighting the three criteria, we generated an ordinal priority scheme in five categories (Fig 4A), indicating those currently considered to be strongholds for the species conservation. The main core areas that currently retain ideal conditions for the species, those in Category 1 thus the most preserved, are in the central and eastern ranges of the Andes, concentrated towards the largest, less disturbed, continuous habitat cores in Colombia. However, several opportunities arise for the following two categories, in which important relicts are still present and under appropriate conservation, and those where potentially restoration schemes could significantly contribute to the maintenance of the species´ populations (Fig 4A). Again, there is a strong relationship between the largest areas with the largest contribution to connectivity and lower levels of human influence (Fig 4B).

## Conservation measures for *Leopardus tigrinus* populations in Colombia

Our estimation of mean number of individuals, based on previous calculations for likely a closely related species, *Leopardus guttulus*, highlight the significant composition of small core habitats, harboring potentially very few individuals. Mean density estimates for the four ranges obtained from previous studies indicated a mean (±SD) density of 0.114±0.051 individuals/km$^2$. The large variation between previous estimates indicates a potentially large variation in the number of individuals present on each core habitat across the country and thus for the whole country (Table 2). According to the areas with potential occurrence identified by our model, the number of Northern Tiger Cat individuals estimated for Colombia is 10,375±4674, but with significant variation (Table 2). Only 12 core habitats (above 1000 km$^2$) contribute over 83% of the total national population (Fig 5A) and almost 95% of the identified potentially cores harbor under 100 individuals (Fig 5B).

Interestingly, in terms of protection level, we found that the species is estimated to be present in at least 415 protected areas from the national to the local level, representing about 33,270.68 km$^2$ (about 36.47% of the total range) (Fig 6A). The majority, albeit with very low coverage (114.62 km$^2$), are represented by Civil Society Natural Reserves (Reservas Naturales de la Sociedad Civil), followed by regional protected areas (Regional Integrated Management Districts—Distritos Regionales de Manejo Integrado; Table 3). National Parks (Parques Nacionales Naturales) were the category with the highest proportion of range covered, however, Unique Natural Areas (Área Natural Única) were the ones that covered the least area (2.98 km$^2$)–which is represented in a single protected area (ANU Los Estoraques; Table 3). It is important to highlight that only 23.71% of the species distribution is covered by dedicated public lands, with mostly strict conservation, while 12.76% is under private lands where the mixed uses is allowed (Table 3). Furthermore, 25.87% of the range is protected by national level protected areas, 10.47% under regional protected areas and only 0.1% under local protected areas.

Of the 183 core habitats, only 92 (50.27%) currently have any level of protection (Fig 6B). Furthermore, mean (±SD) level of protection across cores currently reach 0.51±0.35, while the mean number of protected areas covering a core reached 9.18±28.94. Over time, coverage of remaining core areas has increased significantly since 1938, with notable increases, in terms of number of cores (or fragments of cores) and area, for 1976, 2008 and 2018 (Fig 6C).

Finally, it is important to highlight that only seven protected areas, primarily from the national level (National Parks), but also one from the regional level (Regional Natural Park),

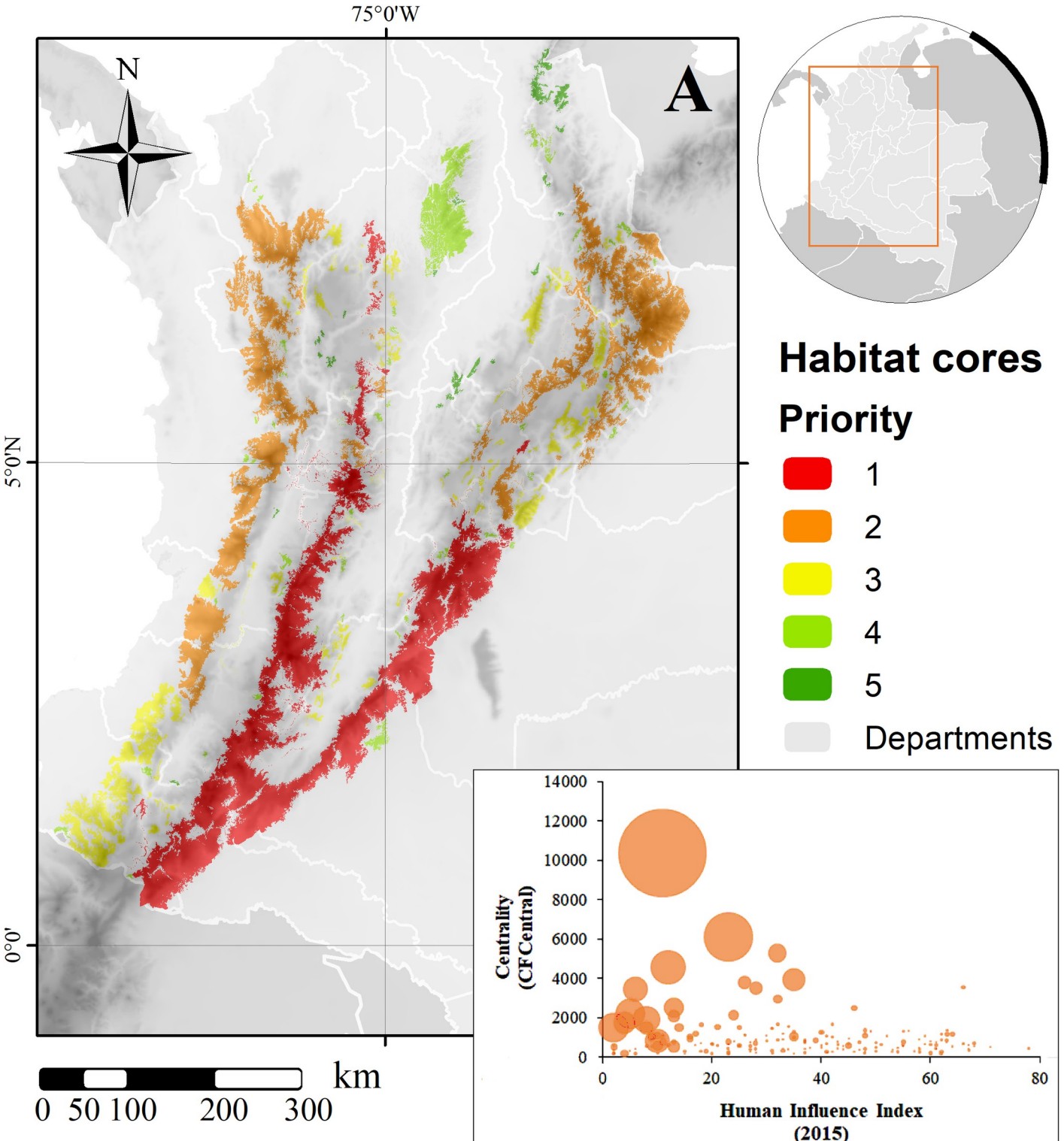

**Fig 4.** (A) Prioritization of remaining habitat cores for *L. tigrinus* in Colombia and (B) relationship between human influence (X-axis), connectivity importance (centrality; Y-axis) and core size (bubble size) of all habitat cores. Background shows topography [79]; map created using ArcGIS [75].

**Table 2. Population density estimation and potential population size, based on previous estimations, in core habitats within the potential distribution of *L. tigrinus* in Colombia.**

| Type of estimation | Range | Mean estimated population | ±SD | Source |
|---|---|---|---|---|
| Typical density | 0.01–0.05 | 2736 | 1824 | [15] |
| High density | 0.1–0.25 | 15962 | 6841 | [15] |
| | 0.15±0.08 | 13681 | 7297 | [76] |
| Fragmented landscape* | 0.07–0.13 | 9121 | 2736 | [15] |
| Mean estimate | 0.114±0.051 | 10375 | 4674 | Our estimation |

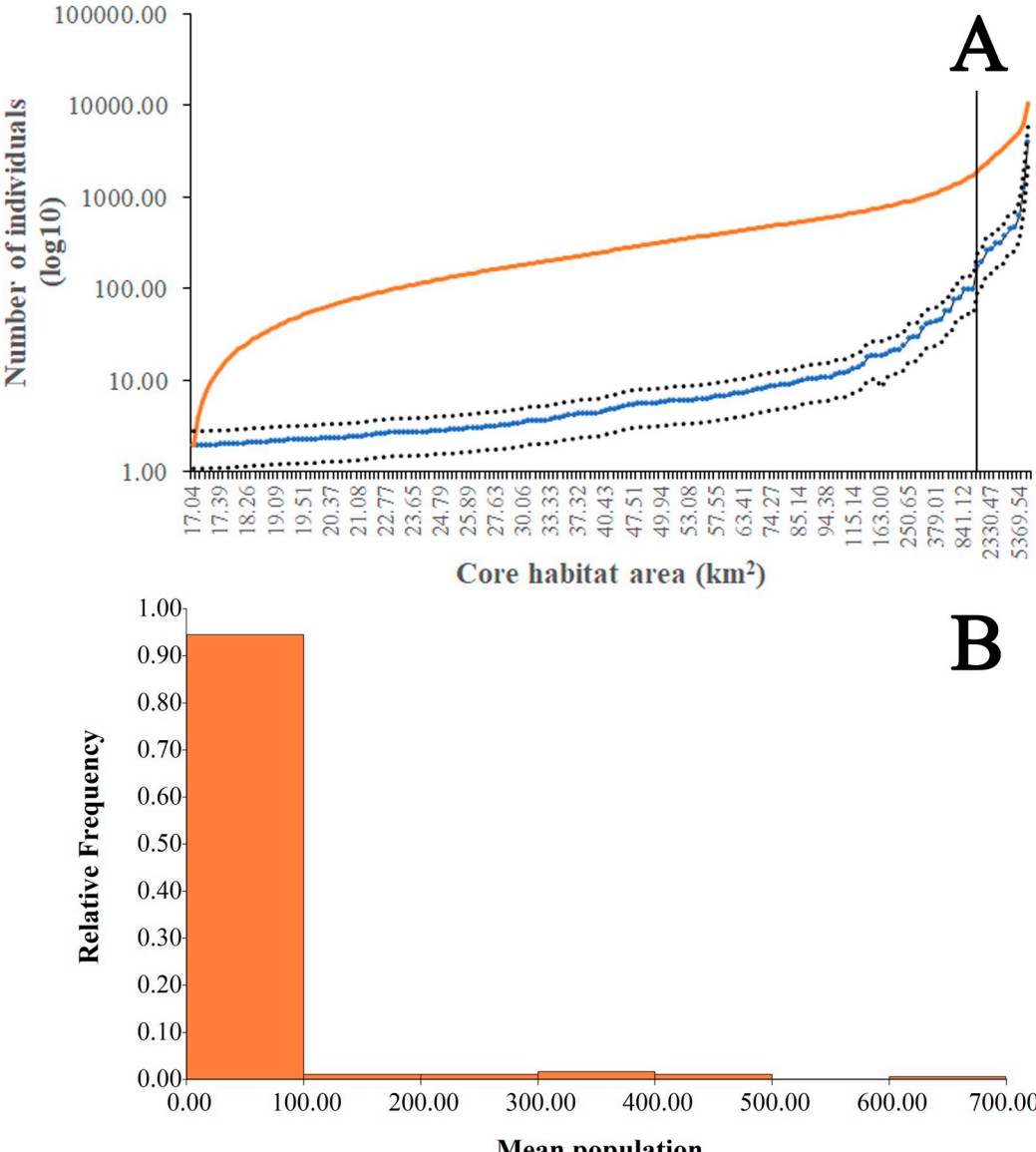

**Fig 5.** (A) Mean (blue line), maximum and minimum (dotted lines), number of potential individuals per core habitat organized by increasing size, and cumulative potential population size (orange line) of *L. tigrinus* present at available habitat cores in Colombia and (B) frequency histogram showing the distribution of the number of core habitats according to the potential population size of *L. tigrinus* in Colombia.

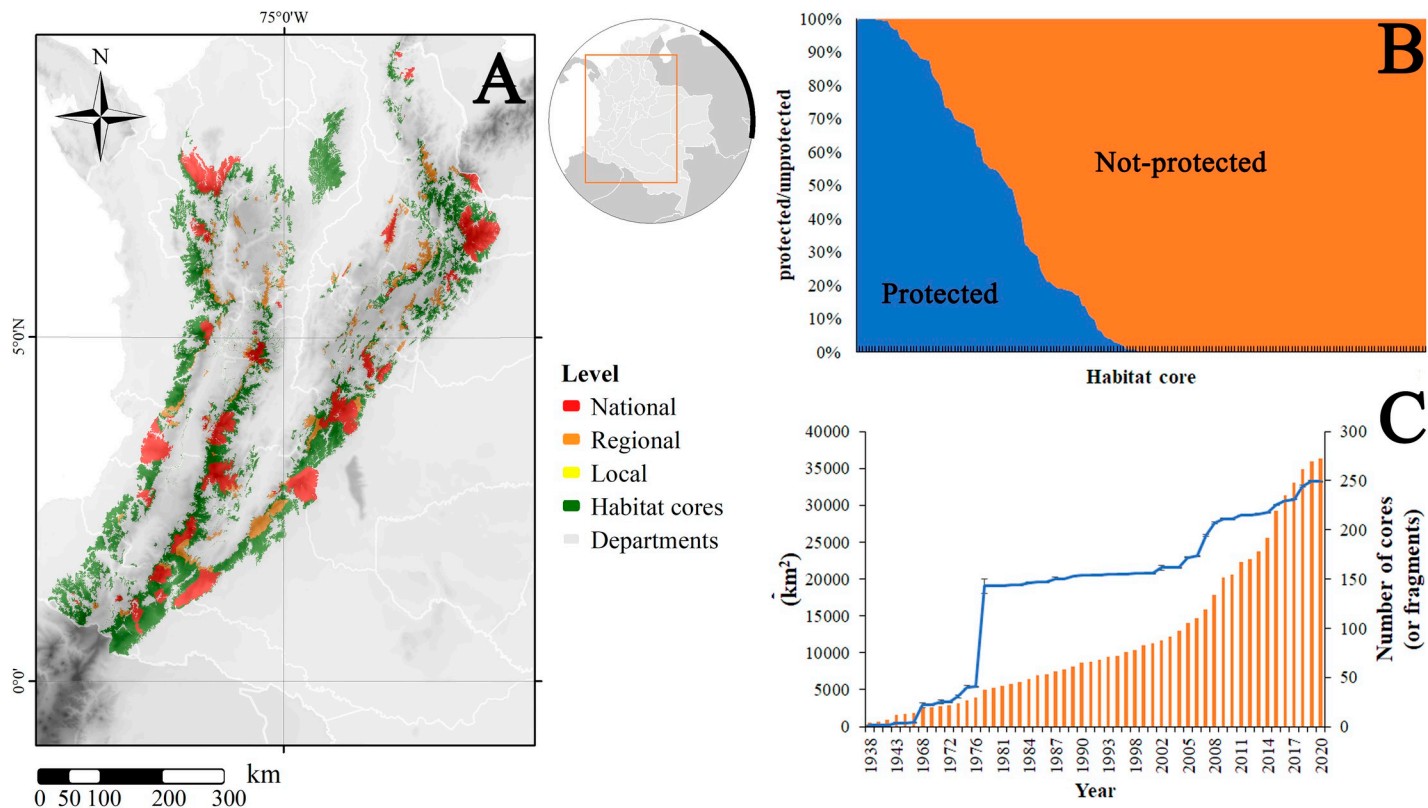

**Fig 6.** (A) Geographic distribution of protected areas currently protecting the potential range of *L. tigrinus* in Colombia, (B) total proportion of protected areas across available habitat cores and (C) temporal evolution of range (line) and habitat cores (bars) protection in Colombia. Background shows topography [79]; map created using ArcGIS [75].

account for almost half (16.76%) of the total range protected. These areas include: El Cocuy (2917.92 km$^2$), Paramillo (2806.59 km$^2$), Sumapaz (2144.15 km$^2$), Cordillera de los Picachos (1918.87 km$^2$), Farallones de Cali (1658.95 km$^2$), Nevado del Huila (1586.69 km$^2$), and Las Hermosas-Gloria Valencia de Castaño (1192.72 km$^2$) National Parks and Miraflores Picachos Regional Natural Park (1064.28 km$^2$).

**Table 3. Total area and number of protected areas according to level and category currently covering the potential distribution range of *L. tigrinus* in Colombia.**

| Level | Category (Original in Spanish) | Protection level | Lands Regime | Area (km$^2$) | Number of areas |
|---|---|---|---|---|---|
| National | Fauna and Flora Sanctuary (Santuario de Fauna y Flora) | Dedicated | Public | 231.17 | 4 |
| | Flora Sanctuary (Santuario de Flora) | Dedicated | Public | 100.86 | 2 |
| | National Forest Protection Reserve (Reservas Forestales Protectoras Nacionales) | Mixed | Private | 1974.69 | 39 |
| | National Park (Parque Nacional Natural) | Dedicated | Public | 21290.29 | 22 |
| | Natural Unique Area (Area Natural Unica) | Dedicated | Public | 2.98 | 1 |
| Regional | Regional Forest Protection Reserve (Reservas Forestales Protectoras Regionales) | Mixed | Private | 1126.98 | 52 |
| | Regional Integrated Management District (Distritos Regionales de Manejo Integrado) | Mixed | Private | 3169.81 | 53 |
| | Regional Natural Park (Parques Naturales Regionales) | Mixed | Private | 5022.62 | 37 |
| | Soil Conservation District (Distritos de Conservacion de Suelos) | Mixed | Private | 236.63 | 6 |
| Local | Civil Society Natural Reserves (Reserva Natural de la Sociedad Civil) | Mixed | Private | 114.63 | 199 |
| **Total** | | | | **33270.68** | **415** |

## Discussion

*Leopardus tigrinus* is among the least known carnivore species in Colombia [14, 24, 79] and even across its distribution its taxonomic status is tenuous at best [16–18]. Considering its presumed widespread distribution, and the long-standing uncertainty regarding its taxonomic position, and given that the form present in the Andes and Central America were until recently considered the same species as *L. guttulus* in the south, and even potentially different between them [18, 80, 81], it remains among the most poorly known felids in South America, with knowledge about the species heterogeneously distributed and overall scarce [17–19, 30]. Despite the recent increased attention to the species and considerable advance in its knowledge and study (e.g., [17]), information about its conservation status is still scarce and mostly unavailable for conservation practices [15, 17, 76].

As previously mentioned, information about the Andean and Central American forms is restricted to sporadic distribution records [14, 20–23, 82, 83], with very few research focused on systematic assessments of its distribution [14, 32] and some limited aspects of its ecology [31, 82, 84, 85]. Such information gaps not only limit the overall knowledge of the species, but also the design of appropriate conservation measures, especially considering that the species is categorized as Vulnerable globally on the IUCN Red List of Threatened Species [19]. Here we present an updated, and systematic assessment of the species distribution for Colombia, with significant considerations relevant for Northern Tiger Cat conservation at the national scale. In addition, our methodological approach can be adopted by other countries to evaluate its current conservation status.

Northern Tiger Cat still retains a widespread distribution at high altitude in Colombia, though largely fragmented and located within the most disturbed areas of the country [65]. Although some very considerable remnants remain, mostly on the Eastern and Central ranges, nearly 93% of its current suitable habitat cores are largely fragmented and isolated, with very small potential populations and under protection. Therefore, based on population estimations and conservation assessments for the species and likely the recently separated *L. guttulus* in Brazil, for instance [15], could indicate, that the Colombian population of *L. tigrinus* is significantly threatened (i.e., from 2,736 individuals using a "typical" density to almost 16,000 individuals on the most optimistic scenario). Especially considering the distribution of the species on the most populated and transformed area of the country [25, 86]. Nevertheless, and considering uncertainty on the actual densities that the species could have in the country, the species could be potentially more threatened than we currently realize, even reaching the critical numbers proposed for the Southern Tiger Cat [87].

It is remarkable however that some large continuous habitats remain across the environmentally suitable areas for the species and represent unique opportunities for designing functional landscapes through complementary conservation measures [88], especially those that consider ecological connectivity on large spatial scales [88–90]. Our prioritization approach identified very considerable continuous habitats towards the eastern and central Andean ranges, but with very promising large, forested areas on the northern portion of the eastern and western branches, where numerous conservation opportunities arise. For instance, the Andean-Orinoco piedmont has been widely recognized as a conservation priority, given its unique biota, and thus, it is likely, and desirable, that it will receive substantial conservation efforts [91–93]. Furthermore, the connectivity network proposed in this work would allow to think on functional connected landscapes especially by articulating marginal fragments to the largest core remaining habitats; this is especially necessary to overcome some of the most significant movement barriers, such as Bogota in the Eastern branch, and reconnecting the northernmost portion of the range in Colombia.

Current coverage of protected areas across the species distribution probably represents the best opportunity for its conservation. However, several considerations arise from the distribution across multiple levels and categories, and especially given the very low overall protected proportion (i.e., around 50% of all cores and all under 50% protected). For instance, only a very low proportion of the estimated range is under strict, public, protected areas, while a significant proportion is under private lands and with very low or absent active management and protection [94–96]. Furthermore, a significant number of cores are represented on very small areas, mostly private reserves (i.e., Reservas de la Sociedad Civil), that represent a very important conservation strategy [86, 97], which by themselves contribute very little to the overall population conservation goals, but act as important stepping stones between main habitat core; such role heavily depends on restoring connectivity across our proposed corridors´ network. Additionally, we highlight the relevance of seven national protected areas (i.e., El Cocuy, Paramillo, Sumapaz, Cordillera de los Picachos, Farallones de Cali, Nevado del Huila and Las Hermosas—Gloria Valencia de Castaño National Parks) and one recently created regional area (i.e., Miraflores Picachos Regional Natural Park), which represent the most important strongholds for the species. The effective maintenance and management of these areas represent a unique opportunity for the species conservation in the long term [65, 96, 98].

It is important to highlight the exploration of the temporal protected areas evolution covering the current species range. The creation of new protected areas during the 1970s and 2000s, and even with the recent expansion of regional and local level areas, significantly increased the coverage, adding new areas relevant for the species. However, the proportion of area covered is not the most effective metric through time, because the number of fragments seems to growth at a faster rate, revealing a trend that needs further refined analyses. This also to better understand how new conservation figures can better contribute to conserving the species either by adding continuous habitats or instead, by increasing the number of smaller fragments protected and connected, likely where functionally and genetically isolated population could survive. Nevertheless, at the current pace of creation of new regional and local protected areas it is likely the total coverage for the species will steadily grow benefiting the overall level of protection, but probably requiring complementary conservation measures that warrant functional landscapes; our connectivity network proposal, although can rapidly change according to land cover changes, probably represent a critical aspect for the functionality and real contribution of this increasing number of protected, yet probably isolated, fragments. Other effective area-based conservation measures (OECMs) articulated with the existing and increasing protected areas can ensure a functional landscapes, both protected and connected, for this species at the long term [99].

Our effort, although with the already known limitations and uncertainty related to SDMs [45, 100, 101], provides a meaningful basic information that could inform and support decision making. The species distribution located in one of the most populated and transformed areas of the country [25, 28, 65] poses an especially complex situation not only for the species conservation, but in general for the Andean biodiversity protection [26, 65], particularly in Colombia [90]. However, recently the species have received special attention given its prominence in the last remaining peri-urban areas on the capital city Bogota [30, 31, 36] and the Aburrá Valley in Antioquia [102]. This unique opportunity has drawn attention for the species, especially in the context of cities that demands more resources and territories for supporting an increasing growing population [103, 104]. For instance, the distribution of the species, although based on very limited assessments [30], has being recently used as a conservation spatial determinant for development and infrastructure projects in the region; thus, providing this information represents a unique opportunity to support and safeguard the last remaining habitats available for the species. Most importantly, it can represent an incredible opportunity for

the investment of compensation funds derived from environmental liabilities of development projects; projects that can support and strengthen different conservation strategies within diverse territories by supporting actions within other effective area-based conservation measures.

Here we present not only an updated distribution assessment for one charismatic and poorly known representative species of the Andes, but also several important arguments to support multiple conservation opportunities in Colombia. Although this assessment warrants its validation based on field efforts, it represents a necessary first step for a unique opportunity to appropriately invest the development resources on safeguarding the last remaining Andean habitats in Colombia.

## Supporting information

**S1 Fig. Map of ecoregions selected for the species distribution modeling of *Leopardus tigrinus* in Colombia.** Background shows topography [79]; map created using ArcGIS [75]. (DOCX)

**S2 Fig. Effect of the predictor variables over *Leopardus tigrinus* distribution model for Colombia.** Bio20: Elevation, Bio4: Temperature seasonality, Bio12: Annual precipitation, Bio14: precipitation of the driest month and Bio02: Mean diurnal range. (DOCX)

**S1 Table. Records used for species distribution modeling of *Leopardus tigrinus* in Colombia classified according to filters for type of evidence, source, and geographic precision.** (DOCX)

**S2 Table. Spearman correlation tests for bioclimatic variables used for the distribution model of *Leopardus tigrinus* in Colombia.** (DOCX)

**S3 Table. Models constructed for species distribution modeling and corresponding parameters for *Leopardus tigrinus* in Colombia.** (DOCX)

## Acknowledgments

This study is part of the ongoing conservation and compensation efforts of Grupo de Energía de Bogotá and ProCAT Colombia for the species in Colombia. Thanks to the staff at GEB and ProCAT for their support throughout the project.

## Author Contributions

**Conceptualization:** José F. González-Maya, Diego A. Zárrate-Charry, Magda Gissella Vargas-Gómez, Teresa Andrea Cárdenas, Victor Mallarino, Jan Schipper.

**Data curation:** José F. González-Maya, Diego A. Zárrate-Charry, Leonardo Lemus-Mejía, Angela P. Hurtado-Moreno.

**Formal analysis:** José F. González-Maya, Diego A. Zárrate-Charry, Leonardo Lemus-Mejía, Angela P. Hurtado-Moreno, Magda Gissella Vargas-Gómez.

**Investigation:** José F. González-Maya, Diego A. Zárrate-Charry, Teresa Andrea Cárdenas.

**Methodology:** José F. González-Maya, Diego A. Zárrate-Charry, Andrés Arias-Alzate, Leonardo Lemus-Mejía.

Writing – original draft: José F. González-Maya, Diego A. Zárrate-Charry, Andrés Arias-Alzate, Leonardo Lemus-Mejía, Angela P. Hurtado-Moreno, Magda Gissella Vargas-Gómez, Teresa Andrea Cárdenas, Victor Mallarino, Jan Schipper.

Writing – review & editing: José F. González-Maya, Diego A. Zárrate-Charry, Andrés Arias-Alzate, Leonardo Lemus-Mejía, Angela P. Hurtado-Moreno, Magda Gissella Vargas-Gómez, Teresa Andrea Cárdenas, Victor Mallarino, Jan Schipper.

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
