## [Decision Letter · Decision Letter 0]

21 Mar 2022

PONE-D-21-29326Spotting what’s important: priority areas, connectivity, and conservation of the Oncilla (Leopardus tigrinus) in Colombia Leopardus tigrinus in ColombiaPLOS ONE

Dear Dr. González-Maya,

Thank you for submitting your manuscript to PLOS ONE. After careful consideration, we feel that it has merit but does not fully meet PLOS ONE’s publication criteria as it currently stands. The invited reviewers have pointed out some critical issues, such as the need to update the current taxonomy of the target species and the need to add a more detailed description of the methods, besides minor editions throughout the text. Please check out their suggestions for different parts of the manuscript, which will certainly improve it. 

We look forward to receiving your revised manuscript.

Kind regards,

Ricardo Bomfim Machado, D.Sc.

Academic Editor

PLOS ONE

Journal Requirements:

3. Thank you for stating the following in the Acknowledgments Section of your manuscript: "This study was funded by Grupo Energía Bogotá S.A E.S.P and ProCAT Colombia and it is part of the ongoing conservation and compensation efforts for the species of both institution in Colombia. "

Please remove any funding-related text from the manuscript and let us know how you would like to update your Funding Statement. Currently, your Funding Statement reads as follows: "This project was jointly funded by ProCAT Colombia and Grupo de Energía de Bogotá S.A E.S.P through mutual agreement. The funders had no role in study design, data collection and analysis, decision to publish, or preparation of the manuscript."

5. We note that Figures 1, 2, 3, 4, and 6 in your submission contain satellite images which may be copyrighted. All PLOS content is published under the Creative Commons Attribution License (CC BY 4.0), which means that the manuscript, images, and Supporting Information files will be freely available online, and any third party is permitted to access, download, copy, distribute, and use these materials in any way, even commercially, with proper attribution. For these reasons, we cannot publish previously copyrighted maps or satellite images created using proprietary data, such as Google software (Google Maps, Street View, and Earth). For more information, see our copyright guidelines: http://journals.plos.org/plosone/s/licenses-and-copyright.

a. You may seek permission from the original copyright holder of Figures 1, 2, 3, 4, and 6 to publish the content specifically under the CC BY 4.0 license.  

Reviewers' comments:

Reviewer's Responses to Questions

**Comments to the Author**

1. Is the manuscript technically sound, and do the data support the conclusions?

Reviewer #1: Yes

Reviewer #2: Yes

2. Has the statistical analysis been performed appropriately and rigorously? 

Reviewer #1: Yes

Reviewer #2: Yes

3. Have the authors made all data underlying the findings in their manuscript fully available?

Reviewer #1: Yes

Reviewer #2: Yes

4. Is the manuscript presented in an intelligible fashion and written in standard English?

Reviewer #1: Yes

Reviewer #2: Yes

5. Review Comments to the Author

Reviewer #1: The manuscript entitled “Spotting what’s important: priority areas, connectivity, and conservation of the Oncilla (Leopardus tigrinus) in Colombia” focuses on assessing the distribution of the species Leopardus tigrinus in Colombia, with the development of distribution models and connectivity analyses to assess the conservation status of habitats identified as priorities for the species. The manuscript presents current and sophisticated analyses for achieving the objectives and is well written overall. However, I consider the main contribution of the paper to be the development of a useful methodology for assessing the habitats and protection status of the species that could be easily incorporated in other areas of its distribution, as well as for several other threatened species. Despite the importance and relevance of the theme of the manuscript, I feel that some changes and elucidations are necessary for the article to progress through the review process. Below are my comments.

Lines 69-78: You need to consider the current taxonomic classification recognized for the “L. tigrinus complex”. Today we have a general recognition of the species Leopardus guttulus in the southern portion of the range of this species’ complex (Trigo et al. 2013, Kitchener et al. 2017, Nascimento & Feijó 2017, de Oliveira, T., Trigo, T., Tortato, M., Paviolo, A., Bianchi, R. & Leite-Pitman, M.R.P. 2016. Leopardus guttulus. The IUCN Red List of Threatened Species 2016). Thus, the information about it being one of the most widely distributed species in the Americas (lines 69-70) and the distribution shown in lines 73-74 are not up to date, or even correct. The lack of consensus on the taxonomy and number of species in the northern part of this species’ complex distribution does exist (see Nascimento & Feijó 2017 and Kitchener et al. 2017), nevertheless both studies recognize the Colombian cat as Leopardus tigrinus. I suggest a revision of these articles and a rewording of this paragraph. I also suggest reviewing the popular name recognized for this species in these bibliographies, oncilla is commonly used for the central American subspecies/population, while northern tigrina or northern tiger cat is most often cited for L. tigrinus as a whole.

Line 82: Review: Brazil has two or three species, depending on the classification assumed, with which one are you comparing the population of Colombia?

Line 122: please, include the time range of the collected records, the information is in the supplementary material but in my opinion it should be included here as well.

Lines 123-131: I think this part of the manuscript including information about filtering, and especially quality control of the data needs to be better explained. What is the three-step filtering? You said that you generated a categorization scheme for each record including 3 categories: credibility, quality and geographic accuracy. Would this be the 3 steps of filtering? I don't see this categorization as a 3 step process because it is not sequential. I would like to see a table or a complete listing of what you considered within each category, because with the incomplete description as it is in the manuscript now (only examples are presented), several doubts arise about this categorization process. As an example: for credibility, information on the type of evidence collected includes examples as footprint and photograph. For quality, a photographic record was exemplified. That is, we have photography in both categories. Still collected specimen and human observation wouldn't fit into evidence type? Moreover, for credibility, two types of information were used: source and evidence, in this case, each received a categorization between low to high? If yes, how was the final category defined?

Lines 132-135: Please better explain this evaluation of the geographic precision. Did you have the geographic coordinates for all records? Here it seemed to me so that the only information evaluated was the location details as department or municipality reported in the source. If all records had coordinates, the evaluation of this less accurate information does not seem very useful to me. I see the importance of this evaluation if the records had variable information of origin, including from just a department of origin (in this case being of low precision) to specific coordinates taken on the collection site (in this case high precision).

Line 133: ...species ranging on the three ranges... would not be three branches? The three described before for the northern part of the mountain range?

Line137: …we then applied a moderate filter… Why did you consider it moderate if records with any category ranked as “low” were excluded?

Lines 143-146: Was this done to create a mask of the area that would be used in the modeling analyses? If yes, I would suggest using “mask/polygon/or sampling area” instead “calibration area”.

Line149: I believe that reference 48 cited here does not include bioclimatic variables in its analyses.

Line155: Please include some references for this statement “…given its potential influence on species distribution models, especially for the Andean region.”

Lines 160-163: You used the software Wallace which includes both Maxent and ENMeval, and you also said that Maxent is the most widely used algorithm, but you did not inform if you used this or both. Please, include this information.

Line 178-184: This paragraph is confusing and with repeated information, here is a suggested text: “Once a binary map was obtained from the best model, we evaluated the different types of vegetation cover included in the suitable areas according to available information on species’ use. All coverage catalogued as natural by the national cartography dataset and with existing evidence of use by the species from literature and the distribution records, was selected within the binary map.”

Lines 184-187: “Once all coverages were extracted, we defined an area threshold for selecting those remaining patches that could be used by the species either as marginal or core habitats, based on a conservative value estimated from the mean home range reported for the species of 17km2.” I don´t understand that. What threshold was this? It was used to define what would be considered as marginal or core habitats? How? Please, explain better how and why this was done.

Line187-188: “We also selected 9 km as a maximum dispersal threshold for the species, also based on previous ecological studies [15, 17, 41, 59].” What exactly was this information used for? Was it used for the identification of core and marginal habitats? Explain.

Lines 189-190: Delete: “The final output included all available natural habitats within the species potential distribution in the country.” Redundant information.

Line255: please, consider to change “used” by “included”.

Line192: Once all available remaining habitats were selected, we designed an ecological connectivity network analysis across the species distribution based on a circuit theory…” replace “selected” by “identified” and “the species distribution” by “recognized patches”.

Lines 196-198: “We defined as core habitats the available habitats previously identified, and generated a resistance layer based on a the most updated available “human footprint index” developed for Colombia [62]”

Change to “We consider as core habitats the recognized habitat patches remaining within the suitable areas identified in the species’ distribution model. The resistance layer was generated based on the available “human footprint index” developed for Colombia.”

Still about the central areas, I would like to see more information about these, especially how many were exactly identified? Couldn't you identify them in a figure? I believe that information about the size and location of these areas is fundamental. In lines 300-301 you say that all but 27 core areas had at least one connection with other areas, so I just understand that you have more than 27, but I don't know how many. Are they the 183 fragments mentioned in line 1*80? Please make this information clearer.

Line 202: delete “cover” from “land use cover”.

Line 204: delete “specific” from “specific species-specific”.

Lines 210-211: The information present in lines 187-188 should be here.

Lines 214-215: Please include a brief description of what the core value and potential current flow would be. Even if you cite papers that refer to these analyses/indexes, your paper should provide enough information so that the reader can understand what these indexes would be and why they were made.

Lines 231-233: Could you present more specifically the value classes considered in each category? I think this would be important so that other people can use the same methodology.

Lines 239-240: Consider that these estimates are all for the currently recognized species Leopardus guttulus. You are working with another species, so a sentence including acknowledgement that these estimates may be quite different for the species in Colombia is essential. I understand that this is the only information available for a closer species, but there should be a sentence highlighting this possible bias in your analyses.

Line 251: “…in terms of total covered area and number of cores or core fragments included.” Delete “cores or”.

Line 255: Our model used 15 ecorregions… Weren’t these ecoregions included before generating the models? In the mask area? Wouldn’t it be better: “Our sampling area included 15 ecoregions”?

Lines 261-265: did you not present the results of the correlation between the variables. Was there a correlation? Which ones were excluded from the analyses?

Lines 261-265: I would like to read or see some information about the behavior of the each variable, e.g.: did elevation have a positive or negative relationship with the probability of species’ occurrence?

Line 270: “climatic suitable areas”, It's not just climatic, is it? Elevation was also used as a variable to run the models, wasn't it?

Lines 271-273: “However, given the huge natural cover transformation processes in the main distribution of the species…”. Consider rephrasing this sentence making it clear that you are referring to transformation due to human activities.

Line 278: replace “after selecting” by “after identifying”. Also delete “climatically”, your model was not based exclusively on climatic variables, right?

Line 282: delete “climatic”

Lines 284-286: “However, most remaining habitat patches are concentrated towards the smaller sizes while very few approach the largest sizes (Fig2B).” change to “Most remaining habitat patches were concentrated towards the smaller size, with very few close to the largest one.”

Lines 304-307: Rephrase as: “Centrality values for all cores also varied considerably, with a mean value of 981.53±1128.85 for the entire network, but with an interesting geographical variation: overall centrality value was closely related with the size of the

core (T=16.40, p<0.001, R2Adj=0.60; Fig.3B).”

Lines 304-313: Without knowing exactly what the central values mean it is difficult to understand how interesting these results are. I believe that including a description of this analysis/index is fundamental to understanding the results.

Lines 323-324: Only in this part of the results do we have an indication of what the centrality values are, that is, the importance of the area/fragment for maintaining connectivity. This information needs to be included in the methodology when this index is presented.

Lines 326-327: Call figure 4B at the end of this sentence.’

Lines 330-331: “The main core areas that currently retain ideal conditions for the species (Category 1) are in the central…”. Indicate here that you are talking only about the first category, the most preserved one.

Lines 343-344: “Our estimation of mean number of individuals, based on previous calculations for a closely related species, Leopardus guttulus, …”. Taking into account that the existing estimates are in areas where we currently recognize as belonging to L. guttulus.

In lines 239-240 you said that density estimates are only available for Brazil, citing references 15 (de Oliveira TG, Tortato MA, Bonjorne de Almeida L, de Campos CB, Beisiegel BM. 2013), 17 (Oliveira-Santos LGR, Graipel ME, Tortato MA, Zucco CA, Cáceres NC, Goulart FVB. 2012) and 59 (Tortato M, de Oliveira T. 2005) and that you used these reports to calculate an expected number of individuals to Colombia.

At Table 1 you presented 4 estimates, citing only two references that do not completely agree with those listed in the methodology (15 and 72). Which estimates and references were actually used? Please verify this.

Also in Table 1, the column for average estimated population refers to density in what area? The total area of Colombia, of the SDM, or the area calculated after the identification of the available natural coverages within your model?

Lines 347-348: “This means a large variation in the number of potential individuals present on each core habitat across the country (Table 1).” Table 1 does not show this, i.e., the variation in the number of potential individuals in each core area. The table shows the values of population estimates that exist in the literature for other areas, am I wrong?

Lines 348-349: “Accordingly, the number of individual oncilla estimated for Colombia is 10,375±4674 individuals…” As I highlighted above, was the area considered for this estimation the total area of Colombia or the areas resulting from your analyses (areas of potential occurrence)? I think it makes much more sense to calculate for the area estimated in the paper from your models.

Lines 350-352: delete “~” from the three values presented here.

Lines 397-406: I suggest reviewing this introductory paragraph with the questions I raised about the recognition of L. guttulus as a distinct species.

Line412-413: In agreement with I cited before, the IUCN red list already recognizes L. guttulus as a distinct species from L. tigrinus.

Lines 421-427: Rewrite considering L. guttulus as a distinct species.

Lines 444-448: Suggested sentence change: “ Furthermore, a significant number of cores are represented on very small areas, mostly private reserves (i.e., Reservas de la Sociedad 446 Civil), that represent a very important conservation strategy [93], which by themselves contribute very little to the overall population conservation goals, but act as important stepping stones between main habitat core.”

Lines 460-464: This sentence is long and confusing, please consider rewriting it.

Lines 464-467: Like corridor implementation? I missed a discussion of ecological corridors between the core areas. These are included in your objectives and you present the development of specific analyses for the identification of these corridors.

Lines 483-489: Very long sentence, consider a division to make the text more fluid.

Supporting Information 1: Please, include at the end of the table the description of the abbreviations in the columns U_Time, U_Credibility, U_Geographic and U_obso. Aslo a brief description of what each one of them would be, especially for U_Time and U_Obso.

Supporting Information 2: This table could be ordered by the AICc value making it easier to follow.

Reviewer #2: Dear editor and authors,

this paper presents important information for the conservation of Leopardus tigrinus in Colombia. I have carefully read the version of the manuscript and found it quite interesting. However, there are some points that need improvement.

Introduction

- Line 65: Add a comma after “in fact”.

- Line 69-78: Although the point of the manuscript is not to make a big review of the tigrina complex, the review presented is missing important references and present some wrong information:

- Lines 75-76: For example, the Brazilian tigrinas were divided in two species by Trigo et al. (2013) - Current Biology, and not by Nascimento and Feijó.

Trigo, T., Schneider, A., Lehugeur, L., Silveira, L., Freitas, T. O., & Eizirik, E. (2013). Molecular data reveal complex hybridization and a cryptic species of Neotropical wild cat. Current Biology, 23(24), 2528-2533.

- Line 75: Still, references 17 and 18 cited in this paragraph refers to studies developed in the Atlantic Forest. This region may not be at such high altitudes as the Andes, but is definitively not known as lowlands, as a great portion of it is located in mountainous regions that can reach more than 2,000 meters. Therefore, I suggest changing these terms

- Line 72: Another important reference missing is Trindade et al. (2021), that reviewed the tigrina complex using genome data.

Trindade, F. J., Rodrigues, M. R., Figueiró, H. V., Li, G., Murphy, W. J., & Eizirik, E. (2021). Genome-wide SNPs clarify a complex radiation and support recognition of an additional cat species. Molecular Biology and Evolution, 38(11), 4987-4991.

- Line 73: may be interesting to cite the IUCN reference too

- Lines 82-83: Be careful with this comparison. About which Brazilian species you are talking (L. tigrinus or L. guttulus)? I suggest remove the comparison and just talk about the species in Colombia.

- Line 105: Add a comma after South America

- Line 123: Exclude “for”

Methods

- Lines 126-135: I think it is important to define, for each attribute, what was considered high, medium and low. For example, records from human observation were included in the final dataset? This information may be added as a table in the supplementary information, or at least refer to the Table S1 in this part of the methods.

- Lines 143-146: What about when the points were only in the border of an ecoregion, in a transition zone, this ecoregion was also used for the calibration? If so, wouldn’t this bias the result, overfitting the data to the ecoregions that have well balanced points across their area? It could be useful to have a map with the ecoregions and the presence records in the supporting information.

- Line 157: Change “base” to “based”

- Lines: 184-187: Please provide a citation for this information.

- Lines 187-189: Authors say they selected 9km as the maximum dispersal threshold based on previous studies, but reading the references cited to this information, I was not able to find any information about the dispersal ability of the species (however, I was not able to find Zarrate-Charry dissertation). These papers talk about population density, home range size and movement within home range, but not dispersal ability. I think that the lack of knowledge on the dispersal ability of the species should be stated more clearly.

- Also, authors used a 9km dispersal distance for the connectivity analysis. However, they considered the home range of the species to be 17km2 and the dispersal ability of a species is usually bigger than the home range (see Bowman et al. 2002). Therefore, although the real dispersal distance of the species may not be far from 9km, test a different value (a bigger one) and compare the results could be very useful, as it may change the connectivity between patches.

- Line 197: remove “a” on “or a the most”

- Lines 208-210: Has this method been used before?

- How was the scale (1-100) determined?

- The transformation of the human footprint into resistance layer was performed using a linear transformation? Please add this information.

- Line 224: change “estimate” to “estimated”

Results

- Results and discussion are a little bit repetitive. I believe it would increase the quality of the paper if they were reviewed.

- Line 270: change “has” for “had”

- Line 273: - Line 270: change “reduces” for “reduced”

- Figure 1 requires more explanation, as there are aspects in the figure that are no specified in the legends. Specifically, it’s important to clarify that the background is an elevation raster, that the yellow lines demonstrate Colombian boundaries and what gray lines are (rivers maybe?). Otherwise, these elements should be removed from the map.

- Lines 282-286 and Figure 2: In this part, authors talk about the small amount of big areas, what can be seen in Figure 2B. However, looking at Figure 2A, we see a great amount of areas in blue colors, indicating their big size. In fact, looking to the map, the amount of area in blue seems to be almost as big as the amount of area in red/brown (the smallest areas). Of curse big patches will occur in smaller numbers than small patches, as they occupy a bigger area, so I think it’s fair to add another graph showing the total amount of area that the different patches sizes occupy, and also talk about this in the text.

- Line 284: change “however” for “thus”

- Table 1: The estimation categories presented are a little confused. It is hard to understand how “high” and “fragments” can be under the same criteria. I think a more elaborated explanation is required for this column.

- Figure 5: The caption and the legends of both figures are not very clear. It could be better elaborated.

- Figure 6: change “total proportion protected areas across...” for “total proportion of protected areas across…”

Discussion

- Line 400: citation - Add Trindade et al. (2021)

- Line 403: “lowlands of Brazil” - Again, I would remove this state or change the language to something like "lower elevation areas than the Andes"

- Lines 424-425: Authors state that on the most optimistic scenario, they estimated that there are less than 10,000 individuals. However, on table 1, of the 5 scenarios, 3 estimate more than 10,000 individuals and 1 scenario estimate almost this number (9,000 individuals). It should be less than 10,000 on the LESS optimistic scenario, no?

- Still, according to the IUCN, most of these estimations are bigger than the estimated number of L. guttulus for its entire range, and both L. guttulus and L. tigrinus are listed as vulnerable – just a comparison.

- Line 479: remove “a” cities

6. PLOS authors have the option to publish the peer review history of their article (what does this mean?). If published, this will include your full peer review and any attached files.

Reviewer #1: No

Reviewer #2: No

---

## [Author Response · Author response to Decision Letter 0]

12 Jul 2022

Responses to reviewers

JOURNAL MANAGER NEW NOTE ADDED 07/12/2022:

Dear Freddie Domini,

We added the corresponding credits to all captions in the MS.

Best,

José

Reviewer #1

The manuscript entitled “Spotting what’s important: priority areas, connectivity, and conservation of the Oncilla (Leopardus tigrinus) in Colombia” focuses on assessing the distribution of the species Leopardus tigrinus in Colombia, with the development of distribution models and connectivity analyses to assess the conservation status of habitats identified as priorities for the species. The manuscript presents current and sophisticated analyses for achieving the objectives and is well written overall. However, I consider the main contribution of the paper to be the development of a useful methodology for assessing the habitats and protection status of the species that could be easily incorporated in other areas of its distribution, as well as for several other threatened species. Despite the importance and relevance of the theme of the manuscript, I feel that some changes and elucidations are necessary for the article to progress through the review process. Below are my comments.

R/Thanks for the comments.

Lines 69-78: You need to consider the current taxonomic classification recognized for the “L. tigrinus complex”. Today we have a general recognition of the species Leopardus guttulus in the southern portion of the range of this species’ complex (Trigo et al. 2013, Kitchener et al. 2017, Nascimento & Feijó 2017, de Oliveira, T., Trigo, T., Tortato, M., Paviolo, A., Bianchi, R. & Leite-Pitman, M.R.P. 2016. Leopardus guttulus. The IUCN Red List of Threatened Species 2016). Thus, the information about it being one of the most widely distributed species in the Americas (lines 69-70) and the distribution shown in lines 73-74 are not up to date, or even correct. The lack of consensus on the taxonomy and number of species in the northern part of this species’ complex distribution does exist (see Nascimento & Feijó 2017 and Kitchener et al. 2017), nevertheless both studies recognize the Colombian cat as Leopardus tigrinus. I suggest a revision of these articles and a rewording of this paragraph. I also suggest reviewing the popular name recognized for this species in these bibliographies, oncilla is commonly used for the central American subspecies/population, while northern tigrina or northern tiger cat is most often cited for L. tigrinus as a whole.

R/Thanks for the comment, we modified accordingly. We updated the distribution according to one of the most recent papers and added “disclaimers” that still the discussion keeps going about the taxonomy and thus distribution. Regarding the common name, we followed the suggestion and changed accordingly across the manuscript.

Line 82: Review: Brazil has two or three species, depending on the classification assumed, with which one are you comparing the population of Colombia?

R/Corrected and eliminated the reference (see comment by Rev2).

Line 122: please, include the time range of the collected records, the information is in the supplementary material but in my opinion it should be included here as well.

R/Added.

Lines 123-131: I think this part of the manuscript including information about filtering, and especially quality control of the data needs to be better explained. What is the three-step filtering? You said that you generated a categorization scheme for each record including 3 categories: credibility, quality and geographic accuracy. Would this be the 3 steps of filtering? I don't see this categorization as a 3 step process because it is not sequential. I would like to see a table or a complete listing of what you considered within each category, because with the incomplete description as it is in the manuscript now (only examples are presented), several doubts arise about this categorization process. As an example: for credibility, information on the type of evidence collected includes examples as footprint and photograph. For quality, a photographic record was exemplified. That is, we have photography in both categories. Still collected specimen and human observation wouldn't fit into evidence type? Moreover, for credibility, two types of information were used: source and evidence, in this case, each received a categorization between low to high? If yes, how was the final category defined?

R/Thanks for the comment, and the reviewer was correct that it was confusing. We modified and provided details in the text and in the SI also a suggested by Rev2.

Lines 132-135: Please better explain this evaluation of the geographic precision. Did you have the geographic coordinates for all records? Here it seemed to me so that the only information evaluated was the location details as department or municipality reported in the source. If all records had coordinates, the evaluation of this less accurate information does not seem very useful to me. I see the importance of this evaluation if the records had variable information of origin, including from just a department of origin (in this case being of low precision) to specific coordinates taken on the collection site (in this case high precision).

R/We added a detailed explanation of the procedure. Although it is understandable the comment by the reviewer, regarding why double-checking locality information does not seem useful, we have vast experience with Colombia biodiversity data, where coordinates are often poorly taken or irresponsibly managed, and still, it makes it to GBIF through the Colombian System of Biodiversity Information (many from public institutions with low technical capacity). This way, we secure that these coordinates are precise and accurate; although we might lose some extra records, we make sure the data we used for the model is the best available.

Line 133: ...species ranging on the three ranges... would not be three branches? The three described before for the northern part of the mountain range?

R/Wrong line, but corrected.

Line137: …we then applied a moderate filter… Why did you consider it moderate if records with any category ranked as “low” were excluded?

R/We added an explanation for the decision and a citation.

Lines 143-146: Was this done to create a mask of the area that would be used in the modeling analyses? If yes, I would suggest using “mask/polygon/or sampling area” instead “calibration area”.

R/No, it was specifically for defining the Mobility Area of the species for the modelling; we added a citation where it is detailed.

Line149: I believe that reference 48 cited here does not include bioclimatic variables in its analyses.

R/Thanks, removed.

Line155: Please include some references for this statement “…given its potential influence on species distribution models, especially for the Andean region.”

R/Added.

Lines 160-163: You used the software Wallace which includes both Maxent and ENMeval, and you also said that Maxent is the most widely used algorithm, but you did not inform if you used this or both. Please, include this information.

R/We reworded for clarity. We refer to the algorithm and not the specific software (which is included in Wallace.

Line 178-184: This paragraph is confusing and with repeated information, here is a suggested text: “Once a binary map was obtained from the best model, we evaluated the different types of vegetation cover included in the suitable areas according to available information on species’ use. All coverage catalogued as natural by the national cartography dataset and with existing evidence of use by the species from literature and the distribution records, was selected within the binary map.”

R/Thank you so much for the suggestion. We modified the text using your suggestion.

Lines 184-187: “Once all coverages were extracted, we defined an area threshold for selecting those remaining patches that could be used by the species either as marginal or core habitats, based on a conservative value estimated from the mean home range reported for the species of 17km2.” I don´t understand that. What threshold was this? It was used to define what would be considered as marginal or core habitats? How? Please, explain better how and why this was done.

R/We reworded to make it clearer.

Line187-188: “We also selected 9 km as a maximum dispersal threshold for the species, also based on previous ecological studies [15, 17, 41, 59].” What exactly was this information used for? Was it used for the identification of core and marginal habitats? Explain.

R/We added details.

Lines 189-190: Delete: “The final output included all available natural habitats within the species potential distribution in the country.” Redundant information.

R/Eliminated

Line192: Once all available remaining habitats were selected, we designed an ecological connectivity network analysis across the species distribution based on a circuit theory…” replace “selected” by “identified” and “the species distribution” by “recognized patches”.

R/Changed.

Lines 196-198: “We defined as core habitats the available habitats previously identified, and generated a resistance layer based on a the most updated available “human footprint index” developed for Colombia [62]”

Change to “We consider as core habitats the recognized habitat patches remaining within the suitable areas identified in the species’ distribution model. The resistance layer was generated based on the available “human footprint index” developed for Colombia.”

R/Thanks for the suggestion. Corrected accordingly.

Still about the central areas, I would like to see more information about these, especially how many were exactly identified? Couldn't you identify them in a figure? I believe that information about the size and location of these areas is fundamental. In lines 300-301 you say that all but 27 core areas had at least one connection with other areas, so I just understand that you have more than 27, but I don't know how many. Are they the 183 fragments mentioned in line 1*80? Please make this information clearer.

R/We reworded for clarity. Anyway earlier in the text we mentioned: “After identifying the available natural coverages within these suitable areas, we found the potentially occupied areas to be only 91,209.85 km2 (~39.8% of the SDM), distributed in 183 patches, with a mean (±SD) area of 498.41±2783.58 km2 across the SDM (Fig 2A).” Furthermore, we find this comment odd given that we have those patches represented in Figures 1, 2, 3,4 and 6, characterizing their area, distribution, centrality, priority and overlap with protected areas.

Line 202: delete “cover” from “land use cover”.

R/Corrected.

Line 204: delete “specific” from “specific species-specific”.

R/Corrected.

Lines 210-211: The information present in lines 187-188 should be here.

R/Corrected.

Lines 214-215: Please include a brief description of what the core value and potential current flow would be. Even if you cite papers that refer to these analyses/indexes, your paper should provide enough information so that the reader can understand what these indexes would be and why they were made.

R/We added the corresponding description.

Lines 231-233: Could you present more specifically the value classes considered in each category? I think this would be important so that other people can use the same methodology.

R/We added details to the text.

Lines 239-240: Consider that these estimates are all for the currently recognized species Leopardus guttulus. You are working with another species, so a sentence including acknowledgement that these estimates may be quite different for the species in Colombia is essential. I understand that this is the only information available for a closer species, but there should be a sentence highlighting this possible bias in your analyses.

R/Thanks for the comment, we added the corresponding acknowledgement of the bias.

Line 251: “…in terms of total covered area and number of cores or core fragments included.” Delete “cores or”.

R/Deleted.

Line255: please, consider to change “used” by “included”.

R/Corrected.

Line 255: Our model used 15 ecorregions… Weren’t these ecoregions included before generating the models? In the mask area? Wouldn’t it be better: “Our sampling area included 15 ecoregions”?

R/Corrected and adjusted accordingly.

Lines 261-265: did you not present the results of the correlation between the variables. Was there a correlation? Which ones were excluded from the analyses?

R/We added the details of the correlation tests.

Lines 261-265: I would like to read or see some information about the behavior of the each variable, e.g.: did elevation have a positive or negative relationship with the probability of species’ occurrence?

R/We added details to the text and added the response curves to the SI.

Line 270: “climatic suitable areas”, It's not just climatic, is it? Elevation was also used as a variable to run the models, wasn't it?

R/Changed for environmentally.

Lines 271-273: “However, given the huge natural cover transformation processes in the main distribution of the species…”. Consider rephrasing this sentence making it clear that you are referring to transformation due to human activities.

R/Corrected.

Line 278: replace “after selecting” by “after identifying”. Also delete “climatically”, your model was not based exclusively on climatic variables, right?

R/Changed

Line 282: delete “climatic”

R/Deleted.

Lines 284-286: “However, most remaining habitat patches are concentrated towards the smaller sizes while very few approach the largest sizes (Fig2B).” change to “Most remaining habitat patches were concentrated towards the smaller size, with very few close to the largest one.”

R/Corrected.

Lines 304-307: Rephrase as: “Centrality values for all cores also varied considerably, with a mean value of 981.53±1128.85 for the entire network, but with an interesting geographical variation: overall centrality value was closely related with the size of the

core (T=16.40, p<0.001, R2Adj=0.60; Fig.3B).”

R/Corrected.

Lines 304-313: Without knowing exactly what the central values mean it is difficult to understand how interesting these results are. I believe that including a description of this analysis/index is fundamental to understanding the results.

R/More details were added to the methods section.

Lines 323-324: Only in this part of the results do we have an indication of what the centrality values are, that is, the importance of the area/fragment for maintaining connectivity. This information needs to be included in the methodology when this index is presented.

R/ More details were added to the methods section.

Lines 326-327: Call figure 4B at the end of this sentence.’

R/Added.

Lines 330-331: “The main core areas that currently retain ideal conditions for the species (Category 1) are in the central…”. Indicate here that you are talking only about the first category, the most preserved one.

R/Added

Lines 343-344: “Our estimation of mean number of individuals, based on previous calculations for a closely related species, Leopardus guttulus, …”. Taking into account that the existing estimates are in areas where we currently recognize as belonging to L. guttulus.

R/Corrected.

In lines 239-240 you said that density estimates are only available for Brazil, citing references 15 (de Oliveira TG, Tortato MA, Bonjorne de Almeida L, de Campos CB, Beisiegel BM. 2013), 17 (Oliveira-Santos LGR, Graipel ME, Tortato MA, Zucco CA, Cáceres NC, Goulart FVB. 2012) and 59 (Tortato M, de Oliveira T. 2005) and that you used these reports to calculate an expected number of individuals to Colombia.

At Table 1 you presented 4 estimates, citing only two references that do not completely agree with those listed in the methodology (15 and 72). Which estimates and references were actually used? Please verify this.

Also in Table 1, the column for average estimated population refers to density in what area? The total area of Colombia, of the SDM, or the area calculated after the identification of the available natural coverages within your model?

R/Thanks for the comment and for pointing out our mistake and also we reworded the caption respond to the last remark. We changed the text and the caption of the table to make sure it is clear that we did all our estimations for the core habitats located within the potential distribution (not the entire territory of Colombia, although it is stated from the methods section). Also, we added the missing citation to the methods section; we also clarify that we used all the available data to select the highest, typicial and fragmented estimates and then the mean estimate using all these studies to make our calculations, thus the use of all these sources although not cited in the table.

Lines 347-348: “This means a large variation in the number of potential individuals present on each core habitat across the country (Table 1).” Table 1 does not show this, i.e., the variation in the number of potential individuals in each core area. The table shows the values of population estimates that exist in the literature for other areas, am I wrong?

R/Correct! Thanks for pointing that out. We have a Figure afterwards to show the variation but reworded for clarity before table1. Also we added a detailed explanation in the methods.

Lines 348-349: “Accordingly, the number of individual oncilla estimated for Colombia is 10,375±4674 individuals…” As I highlighted above, was the area considered for this estimation the total area of Colombia or the areas resulting from your analyses (areas of potential occurrence)? I think it makes much more sense to calculate for the area estimated in the paper from your models.

R/Absolutely right, modified the phrase to make it clearer and added details to the methods.

Lines 350-352: delete “~” from the three values presented here.

R/Corrected.

Lines 397-406: I suggest reviewing this introductory paragraph with the questions I raised about the recognition of L. guttulus as a distinct species.

R/Agreed and corrected.

Line412-413: In agreement with I cited before, the IUCN red list already recognizes L. guttulus as a distinct species from L. tigrinus.

R/We checked the paragraph but we believe there is no reason to call L. guttulus assessment. This paragraph only says that we don´t now much, about the species, that is catalogued as Vulnerable by IUCN (the same year L. guttulus was assessed and by the same author) and that we present new information.

Lines 421-427: Rewrite considering L. guttulus as a distinct species.

R/Changed accordingly.

Lines 444-448: Suggested sentence change: “ Furthermore, a significant number of cores are represented on very small areas, mostly private reserves (i.e., Reservas de la Sociedad 446 Civil), that represent a very important conservation strategy [93], which by themselves contribute very little to the overall population conservation goals, but act as important stepping stones between main habitat core.”

R/Changed as suggested.

Lines 460-464: This sentence is long and confusing, please consider rewriting it.

R/Corrected.

Lines 464-467: Like corridor implementation? I missed a discussion of ecological corridors between the core areas. These are included in your objectives and you present the development of specific analyses for the identification of these corridors.

R/We added some specific lines across the discussion to support the approach.

Lines 483-489: Very long sentence, consider a division to make the text more fluid.

R/Corrected.

Supporting Information 1: Please, include at the end of the table the description of the abbreviations in the columns U_Time, U_Credibility, U_Geographic and U_obso. Aslo a brief description of what each one of them would be, especially for U_Time and U_Obso.

R/Added.

Supporting Information 2: This table could be ordered by the AICc value making it easier to follow.

R/Changed accordingly.

Reviewer #2

Dear editor and authors,

this paper presents important information for the conservation of Leopardus tigrinus in Colombia. I have carefully read the version of the manuscript and found it quite interesting. However, there are some points that need improvement.

R/Thanks for the comments.

Introduction

- Line 65: Add a comma after “in fact”.

R/Added.

- Line 69-78: Although the point of the manuscript is not to make a big review of the tigrina complex, the review presented is missing important references and present some wrong information:

- Lines 75-76: For example, the Brazilian tigrinas were divided in two species by Trigo et al. (2013) - Current Biology, and not by Nascimento and Feijó.

Trigo, T., Schneider, A., Lehugeur, L., Silveira, L., Freitas, T. O., & Eizirik, E. (2013). Molecular data reveal complex hybridization and a cryptic species of Neotropical wild cat. Current Biology, 23(24), 2528-2533.

R/Added.

- Line 75: Still, references 17 and 18 cited in this paragraph refers to studies developed in the Atlantic Forest. This region may not be at such high altitudes as the Andes, but is definitively not known as lowlands, as a great portion of it is located in mountainous regions that can reach more than 2,000 meters. Therefore, I suggest changing these terms

R/Changed.

- Line 72: Another important reference missing is Trindade et al. (2021), that reviewed the tigrina complex using genome data.

Trindade, F. J., Rodrigues, M. R., Figueiró, H. V., Li, G., Murphy, W. J., & Eizirik, E. (2021). Genome-wide SNPs clarify a complex radiation and support recognition of an additional cat species. Molecular Biology and Evolution, 38(11), 4987-4991.

R/Thanks for the suggestion, we added the reference.

- Line 73: may be interesting to cite the IUCN reference too

R/Added.

- Lines 82-83: Be careful with this comparison. About which Brazilian species you are talking (L. tigrinus or L. guttulus)? I suggest remove the comparison and just talk about the species in Colombia.

R/Agree and changed.

- Line 105: Add a comma after South America

R/Added.

- Line 123: Exclude “for”

R/Corrected.

Methods

- Lines 126-135: I think it is important to define, for each attribute, what was considered high, medium and low. For example, records from human observation were included in the final dataset? This information may be added as a table in the supplementary information, or at least refer to the Table S1 in this part of the methods.

R/Thanks, we added details and the table.

- Lines 143-146: What about when the points were only in the border of an ecoregion, in a transition zone, this ecoregion was also used for the calibration? If so, wouldn’t this bias the result, overfitting the data to the ecoregions that have well balanced points across their area? It could be useful to have a map with the ecoregions and the presence records in the supporting information.

R/Thanks for the comment, we added some details in methods and results, and also added the map in SI.

- Line 157: Change “base” to “based”

R/Corrected.

- Lines: 184-187: Please provide a citation for this information.

R/Added.

- Lines 187-189: Authors say they selected 9km as the maximum dispersal threshold based on previous studies, but reading the references cited to this information, I was not able to find any information about the dispersal ability of the species (however, I was not able to find Zarrate-Charry dissertation). These papers talk about population density, home range size and movement within home range, but not dispersal ability. I think that the lack of knowledge on the dispersal ability of the species should be stated more clearly.

- Also, authors used a 9km dispersal distance for the connectivity analysis. However, they considered the home range of the species to be 17km2 and the dispersal ability of a species is usually bigger than the home range (see Bowman et al. 2002). Therefore, although the real dispersal distance of the species may not be far from 9km, test a different value (a bigger one) and compare the results could be very useful, as it may change the connectivity between patches.

R/Thanks for the comment and we totally agree it was confusing. We provided extra details on why we used different figures trying to be as conservative as possible given the inherent uncertainty from using ecological attributes from a different species. Details were provided both on the selection of core habitats and the connectivity analyses. We expect the details are now clearer specifically on how we made the methodological decisions.

Indeed, the studies we found in our review did not present specific information regarding dispersal distance. The studies that we found were mainly directed to understanding the home range of the species. We used such information to calculate a proxy for its dispersal distance based on the proposal of Bowman et al. 2002. To define the home range to calculate the median dispersal distance we used as threshold 9 km, which represents a dispersal distance of an individual with a home range of 4.5 km 2 , which represent the median value of all reported home ranges for the species. We used a median value based on the distribution of the existing values. We maintained a mean value for the dispersal distance to reduce potential number of corridors of length that exceed its use within conservation planning strategies. The identification of corridors using circuit theory provides landscape solutions that require efforts to become functional in the territories. 

We did not used within our analysis a dispersal distance based on the same home range used to define minimum core habitat patch size. We used different values to create the spatial representation. To define core habitat patches we used the maximum home range reported for the species (17km 2 ). We use this threshold as a prioritization measure, to ensure that the remaining patches can have at least one individual. We did not used a conservative approach and maintain patches smaller to 17km 2 , due to the inherent uncertainty of such patches to maintain at least one individual. The higher value encloses for sure all the existing reported home ranges. 

- Line 197: remove “a” on “or a the most”

R/Corrected.

- Lines 208-210: Has this method been used before?

- How was the scale (1-100) determined?

- The transformation of the human footprint into resistance layer was performed using a linear transformation? Please add this information.

R/We added details and a citation.

- Line 224: change “estimate” to “estimated”

R/Corrected.

Results

- Results and discussion are a little bit repetitive. I believe it would increase the quality of the paper if they were reviewed.

R/We reviewed ancd changed what was possible after following all the rest of the comments. Although we appreciate the comment, we believe the Results section is entirely focused on presenting the results while we try to focus on the conservation aspect on the discussion.

- Line 270: change “has” for “had”

R/Changed.

- Line 273: - Line 270: change “reduces” for “reduced”

R/Changed

- Figure 1 requires more explanation, as there are aspects in the figure that are no specified in the legends. Specifically, it’s important to clarify that the background is an elevation raster, that the yellow lines demonstrate Colombian boundaries and what gray lines are (rivers maybe?). Otherwise, these elements should be removed from the map.

R/Thanks for the comments we modified the figure according to PLOS policies and added details to the caption.

- Lines 282-286 and Figure 2: In this part, authors talk about the small amount of big areas, what can be seen in Figure 2B. However, looking at Figure 2A, we see a great amount of areas in blue colors, indicating their big size. In fact, looking to the map, the amount of area in blue seems to be almost as big as the amount of area in red/brown (the smallest areas). Of curse big patches will occur in smaller numbers than small patches, as they occupy a bigger area, so I think it’s fair to add another graph showing the total amount of area that the different patches sizes occupy, and also talk about this in the text.

R/Thanks for the comment but we believe the reviewer was confused. First, there is only 1 patch in blue which covers over 35000 km, thus is not a lot. Nevertheless, we changed the colors to allow easier reading and we believe the graph and text already clarifies that only 1 patch is really big, the rest are small. Furthermore, the text includes the total area available, the distribution of frequencies and how many patches cover how much of the available habitats. The figure suggested is precisely the figure included in Fig2B (the size of the different patches); in fact, it is not clear what graph could show the amount of area occupied by different patch size (same variable). Nevertheless, we change the map and added some small details in the text.

- Line 284: change “however” for “thus”

R/Followed suggestion by Rev1.

- Table 1: The estimation categories presented are a little confused. It is hard to understand how “high” and “fragments” can be under the same criteria. I think a more elaborated explanation is required for this column.

R/Thanks for pointing that out. We added an explanation in the methods section.

- Figure 5: The caption and the legends of both figures are not very clear. It could be better elaborated.

R/Thanks, we reworded both.

- Figure 6: change “total proportion protected areas across...” for “total proportion of protected areas across…”

R/Changed.

Discussion

- Line 400: citation - Add Trindade et al. (2021)

R/Added.

- Line 403: “lowlands of Brazil” - Again, I would remove this state or change the language to something like "lower elevation areas than the Andes"

R/Eliminated by suggestion of the other reviewer.

- Lines 424-425: Authors state that on the most optimistic scenario, they estimated that there are less than 10,000 individuals. However, on table 1, of the 5 scenarios, 3 estimate more than 10,000 individuals and 1 scenario estimate almost this number (9,000 individuals). It should be less than 10,000 on the LESS optimistic scenario, no?

- Still, according to the IUCN, most of these estimations are bigger than the estimated number of L. guttulus for its entire range, and both L. guttulus and L. tigrinus are listed as vulnerable – just a comparison.

R/Agree, we reworded accordingly and even added the proper comparison with L. guttulus.

- Line 479: remove “a” cities

R/Removed.

---

## [Decision Letter · Decision Letter 1]

8 Aug 2022

PONE-D-21-29326R1Spotting what’s important: priority areas, connectivity, and conservation of the Northern Tiger Cat (Leopardus tigrinus) in ColombiaPLOS ONE

Dear Dr. González-Maya,

Thank you for submitting your manuscript to PLOS ONE. After careful consideration, we feel that it has merit but does not fully meet PLOS ONE’s publication criteria as it currently stands. Therefore, we invite you to submit a revised version of the manuscript that addresses the points raised during the review process.

We look forward to receiving your revised manuscript.

Kind regards,

Bi-Song Yue, Ph.D

Academic Editor

PLOS ONE

Journal Requirements:

Reviewers' comments:

Reviewer's Responses to Questions

**Comments to the Author**

1. If the authors have adequately addressed your comments raised in a previous round of review and you feel that this manuscript is now acceptable for publication, you may indicate that here to bypass the “Comments to the Author” section, enter your conflict of interest statement in the “Confidential to Editor” section, and submit your "Accept" recommendation.

Reviewer #1: All comments have been addressed

Reviewer #2: (No Response)

2. Is the manuscript technically sound, and do the data support the conclusions?

Reviewer #1: Yes

Reviewer #2: Yes

3. Has the statistical analysis been performed appropriately and rigorously? 

Reviewer #1: Yes

Reviewer #2: Yes

4. Have the authors made all data underlying the findings in their manuscript fully available?

Reviewer #1: Yes

Reviewer #2: Yes

5. Is the manuscript presented in an intelligible fashion and written in standard English?

Reviewer #1: Yes

Reviewer #2: Yes

6. Review Comments to the Author

Reviewer #1: Dear Editor, I believe that the authors have made a good review, appropriately answering all my comments and making the necessary changes and corrections. In my opinion, the manuscript is now suitable to be accepted for publication in PlosOne. I am satisfied with the review performed, having only pointed out a few minor text corrections in the article submission pdf itself. My comments are on the version with track changes. I do not need to see a revised version. Best wishes.

Reviewer #2: Dear Editor,

I believe the quality of the manuscript has greatly increased, but there are some points that still require attention:

Lines 75-76 – I would add “central and northern Brazil” as according to the IUCN, L. tigrinus’ range encompasses Brazil; and is not clear if the species from Costa Rica (identified by Trindade et al. 2021) is the same or different one from the one found in Colombia.

Line 221 – Add “potential” dispersal distance.

Line 228 – change “though” to “through”.

Line 290-292 – change “Among the most important predicting variables, elevation (64.3%) and temperature seasonality (27.9%), accounting as the most important according to permutation importance” to “Among the selected variables, elevation (64.3%) and temperature seasonality (27.9%) accounted as the most important according to permutation importance”.

Figure 1 – Figure 1A is a bit confusing. According to the legend, only the lighter green is SDM, so what is the darker green in figure A? It seems the darker green is the habitat cores, but the figure caption states that figure A only demonstrates the species records and the SDM. If it is the habitat cores, I do not see a reason for the existence of figure B. Please clarify.

Line 323 – I believe the 83% is related to the habitat cores and not the SDM, as previously it is stated that the potentially occupied areas represent ~40% of the SDM. Therefore, it would be good to clarify it, maybe changing “83% of the species distribution” to “83% of the identified core areas”, as species distribution may also be related to the SDM.

Line 335 – Here says that the maximum dispersal distance used was 9km, but on the methods (line 219) says it was 15km. Please clarify.

Figure 6 – the colors of figure 6A are hard to distinguish. It may be due to lower resolution of the figure attached in the file. Perhaps it would be interesting to change the colors or try to use polygons or lines overlap to make it easier for the reader.

Line 450 – change “a different” to “the same” or “was until recently considered” to “was recently recognized as different species”. The Andean and Costa Rican tigrina were until recently considered the same species from L. guttulus. They were separated in two species in 2013 (Trigo et al.). In addition, Trindade et al. 2021 showed that the Costa Rican tigrina is also a separate species from the L. tigrinus that occurs in the north and northeastern Brazil.

Line 480 – remove “same”.

Line 495 – I suggest removing “and South America” or rephrasing this sentence.

Line 526 – add “the”: is likely that “the” total coverage for the species…

Line 529 – remove “it”.

7. PLOS authors have the option to publish the peer review history of their article (what does this mean?). If published, this will include your full peer review and any attached files.

Reviewer #1: No

Reviewer #2: No

---

## [Author Response · Author response to Decision Letter 1]

11 Aug 2022

Responses to reviewers

Reviewer #1

Reviewer #1: Dear Editor, I believe that the authors have made a good review, appropriately answering all my comments and making the necessary changes and corrections. In my opinion, the manuscript is now suitable to be accepted for publication in PlosOne. I am satisfied with the review performed, having only pointed out a few minor text corrections in the article submission pdf itself. My comments are on the version with track changes. I do not need to see a revised version. Best wishes.

R/Thanks for the comments, we really appreciate the feedback and constructive criticism!

Eliminate: Mostly known from Brazil (currently suggested as two distinct species)

R/Changed.

Since you are talking now only about the northern tiger cat, this part of the introduction makes no more sense. The cited works (20 and 21) were done in regions of L. guttulus occurrence and if we follow Nascimento & Feijó's classification, we will have L. tigrinus with a restricted distribution in Brazil and as far as I know, we have no studies with these populations. I think it is better to avoid comparative texts with Brazilian populations mainly due to taxonomic uncertainty of tigrinus/emiliae in Brazil.

Also, Nascimento & Feijó split it into 3 species in Brazil: L. tigrinus, L. emiliae and L. guttulus

R/Agreed. We eliminated the phrase and changed for readability.

three criteria: source, evidence and geographic precision.

R/Changed.

Text suggestion: 

Each criterion showed different attributes with an assigned reliability category (high, medium, low) that were latter....

R/Changed.

criteria as in previous studies

R/Corrected.

Erase Three and Criteria

R/Corrected.

You need to standardize the terms used here, from what I understand from the text, "Category" title of the first column would be "Criteria", "Evidence" in the second column would be Attributes

R/Corrected.

Geographic precision

R/Corrected

Brazilian tiger cats

R/Corrected.

(Southern Tiger Cat)

R/Added.

whole country

R/Corrected.

According to the areas with potential occurrence identified by our model, the number of Northern Tiger Cat individuals estimated for Colombia is...

R/Changed.

Southern Tiger Cat

R/Changed.

Reviewer #2

Dear Editor,

I believe the quality of the manuscript has greatly increased, but there are some points that still require attention:

R/ Thank you so much for all the revisions and comments!

Lines 75-76 – I would add “central and northern Brazil” as according to the IUCN, L. tigrinus’ range encompasses Brazil; and is not clear if the species from Costa Rica (identified by Trindade et al. 2021) is the same or different one from the one found in Colombia.

R/We changed to reflect that the Costa Rica form could change, but given the evidence from recent papers and the comments from the other reviewer, we didn’t include Brazil.

Line 221 – Add “potential” dispersal distance.

R/Corrected.

Line 228 – change “though” to “through”.

R/Corrected.

Line 290-292 – change “Among the most important predicting variables, elevation (64.3%) and temperature seasonality (27.9%), accounting as the most important according to permutation importance” to “Among the selected variables, elevation (64.3%) and temperature seasonality (27.9%) accounted as the most important according to permutation importance”.

R/Corrected.

Figure 1 – Figure 1A is a bit confusing. According to the legend, only the lighter green is SDM, so what is the darker green in figure A? It seems the darker green is the habitat cores, but the figure caption states that figure A only demonstrates the species records and the SDM. If it is the habitat cores, I do not see a reason for the existence of figure B. Please clarify.

R/Absolutely right! We changed and just left one map.

Line 323 – I believe the 83% is related to the habitat cores and not the SDM, as previously it is stated that the potentially occupied areas represent ~40% of the SDM. Therefore, it would be good to clarify it, maybe changing “83% of the species distribution” to “83% of the identified core areas”, as species distribution may also be related to the SDM.

R/Corrected.

Line 335 – Here says that the maximum dispersal distance used was 9km, but on the methods (line 219) says it was 15km. Please clarify.

R/Corrected.

Figure 6 – the colors of figure 6A are hard to distinguish. It may be due to lower resolution of the figure attached in the file. Perhaps it would be interesting to change the colors or try to use polygons or lines overlap to make it easier for the reader.

R/We changed the colors; we do think is a matter of resolution. Shadows, given the complexity of shapes, just looks more confusing and messier.

Line 450 – change “a different” to “the same” or “was until recently considered” to “was recently recognized as different species”. The Andean and Costa Rican tigrina were until recently considered the same species from L. guttulus. They were separated in two species in 2013 (Trigo et al.). In addition, Trindade et al. 2021 showed that the Costa Rican tigrina is also a separate species from the L. tigrinus that occurs in the north and northeastern Brazil.

R/Changed to reflect both aspects.

Line 480 – remove “same”.

R/Corrected.

Line 495 – I suggest removing “and South America” or rephrasing this sentence.

R/Removed.

Line 526 – add “the”: is likely that “the” total coverage for the species…

R/Corrected.

Line 529 – remove “it”.

R/Removed.

---

## [Editor Report · Decision Letter 2]

16 Aug 2022

Spotting what’s important: priority areas, connectivity, and conservation of the Northern Tiger Cat (Leopardus tigrinus) in Colombia

PONE-D-21-29326R2

Dear Dr. González-Maya,

We’re pleased to inform you that your manuscript has been judged scientifically suitable for publication and will be formally accepted for publication once it meets all outstanding technical requirements.

Kind regards,

Bi-Song Yue, Ph.D

Academic Editor

PLOS ONE

---

## [Editor Report · Acceptance letter]

2 Sep 2022

PONE-D-21-29326R2 

*Spot*ting what’s important: priority areas, connectivity, and conservation of the Northern Tiger Cat (*Leopardus tigrinus*) in Colombia 

Dear Dr. González-Maya:

I'm pleased to inform you that your manuscript has been deemed suitable for publication in PLOS ONE. Congratulations! Your manuscript is now with our production department. 

Kind regards, 

on behalf of

Dr. Bi-Song Yue 

Academic Editor

PLOS ONE